# Atmospheric and Watershed Modelling of Trifluoroacetic Acid from Oxidation of HFO-1234ze(E) Released by Prospective Pressurized Metered-Dose Inhaler Use

Shivendra G. Tewari[1], Krish Vijayaraghavan[2], Kun Zhao[2], Liji M. David[2], Katie Tuite[2], Felix

Kristanovich[2], Yuan Zhuang[2], Benjamin Yang[2], Cecilia Hurtado[2], Dimitrios K. Papanastasiou[3], Paul

Giffen[4], Holly Kimko[1], Megan Gibbs[1], Stefan Platz[5]

[1]Clinical Pharmacology & Quantitative Pharmacology, R&D BioPharmaceuticals, AstraZeneca, Gaithersburg, USA
[2]Ramboll, Novato, USA
[3]Honeywell, Buffalo, USA
[4]Clinical Pharmacology & Safety Sciences, R&D BioPharmaceuticals, AstraZeneca, Cambridge, UK
[5]Clinical Pharmacology & Safety Sciences, R&D BioPharmaceuticals, AstraZeneca, Baar, Switzerland

*Correspondence to*: Shivendra G. Tewari (shivendra.tewari@astrazeneca.com)

**Abstract.** Trans-1,3,3,3-tetrafluoroprop-1-ene  (E-CF$_3$CH=CHF or HFO-1234ze(E)), is a hydrofluoroolefin (HFO) with near-zero global warming potential, developed as a propellant for use in pressurised metered-dose inhalers (pMDIs). HFO-1234ze(E) has a '–CF$_3$' moiety, which makes formation of trifluoroacetic acid (TFA) possible in the atmosphere. To quantify the contribution of TFA formed from HFO-1234ze(E)-based pMDIs, we performed an extensive study using a global atmospheric model coupled with detailed watershed modelling. Our approach incorporated the master chemical mechanism of HFO-1234ze(E), considering all potential atmospheric degradation pathways, and assumed pMDIs as the sole emission source of HFO-1234ze(E). Based on global pMDI volume sales data, we estimate HFO-1234ze(E) emission rates to be 4.736 Gg yr$^{-1}$. Although emissions are higher in northern-temperate regions, our model predicts that the highest TFA deposition rates occur in the tropics, likely due to more intensive photolysis of trifluoroacetic aldehyde in temperate zones, favouring non-TFA products, and/or transport of TFA into the tropical zone from nearby regions. We then applied a fate-and-transport model to estimate TFA concentrations in surface water, soil, and sediments using predicted TFA deposition around the Hudson, Cauvery, and Rhine River basins over 30 years. Modelled surface water TFA levels ranged between 0.8–19.3 ng L$^{-1}$. Compared to the conservative Netherlands drinking water thresholds (2,200 ng L$^{-1}$), this represents a margin-of-exposure up to 2,500-fold (Rhine) and no lower than 100-fold (Cauvery). These findings suggest that, for the regions assessed, TFA formation resulting from pMDI-related HFO-1234ze(E) emissions is minimal and does not present a risk to human health or the environment.

## 1 Introduction

Asthma and Chronic obstructive pulmonary disease (COPD) are potentially life-threatening diseases, affecting roughly half a billion people globally and cause nearly 4 million deaths each year (Collaborators, 2020). Patients living with respiratory diseases generally rely on pressurized metered dose inhalers (pMDIs), dry powder inhalers, or soft-mist inhalers for treatment

of these conditions. Of these options, pMDIs are the most prescribed devices globally with greater than 90% usage in some countries (Bell et al., 2023). pMDI devices rely on a medical propellant for delivering the active pharmaceutical ingredient (API) to a patient's lungs. Present-day pMDI devices use hydrofluorocarbons (HFCs) as the medical propellant; however, all HFC-based propellants have high global warming potential (GWP), e.g., HFC-134a has a GWP (100-year time horizon) that is 1,470 times greater than $CO_2$ (Organization, 2022), and are subject to phase down under the Kigali amendment of the Montreal Protocol (Heath, 2017). The American Innovation and Manufacturing (AIM) Act, enacted by Congress in 2020, authorized US Environmental Protection Agency (EPA) to phase down the production and consumption of hydrofluorocarbons, such as HFCs, by 2036 (Logan, 2021). Similarly, the European F-gas regulation aims to reduce the supply of HFCs to 3% of the original baseline by 2050 (Union, 2024).

To ensure supply of these essential medications to patients, the pharmaceutical industry in collaboration with Honeywell has developed a near-zero GWP medical propellant (Tewari et al., 2023), trans-1,3,3,3-tetrafluoroprop-1-ene (the E-isomer), correctly written as $E-CF_3CH=CHF$ or HFO-1234ze(E) or next-generation medical propellant (NGP) (a hydrofluoroolefin; HFO), with initial pMDI devices expected to transition this year and transition of all pMDI devices anticipated by 2030 The direct GWP (100-year time horizon) of HFO-1234ze(E) is 1 (Organization, 2022). McGillen et al. (2023) reported a small yield (~3%) of trifluoromethane (HFC-23; $CHF_3$) from its atmospheric degradation, suggesting the GWP could increase to about 12 on a 100-year horizon. However, their model used temperature-independent rate coefficients, even though these reactions are known to have strong temperature dependence. For example, the ozone reaction rate with cis-dichloroethene—a compound with similar reactivity to HFO-1234ze(E)—decreases by a factor of about 100 between 298 K and 220 K (Leather et al., 2011). Accounting for this temperature dependence would further reduce the estimated loss of HFO-1234ze(E) via ozone reaction by McGillen et al. (2023), and thus lower both the predicted global yield of HFC-23 and the indirect GWP of HFO-1234ze(E).

According to the Organization for Economic Cooperation and Development (OECD) definition, HFO-1234ze(E) belongs to a group of synthetic chemicals called per- and polyfluoroalkyl substances (PFAS) which are typically resistant to environmental breakdown. PFAS are a group of nearly 15,000 chemicals (Williams et al., 2017) with substantially diverse physicochemical properties, e.g., HFO-1234ze(E) upon release in the atmosphere breaks down within ~20 days (Tewari et al., 2023; Neale et al., 2021) while some PFAS can take many years to decompose. Despite the diverse structure and behaviour of PFAS chemicals, in 2023, five European Union (EU) countries submitted a blanket-restriction proposal under the Registration, Evaluation, Authorisation and Restriction of Chemicals (REACH) banning all manufacturing and usage of PFAS in the EU with exceptions for human and veterinary APIs, biocides, pesticides, and a few fully degradable PFAS subgroups.

HFO-1234ze(E) falls under the purview of the PFAS-restriction proposal because its structure contains a trifluoromethyl group (–$CF_3$) which makes formation of trifluoroacetic acid (TFA) possible in the atmosphere; TFA is the most abundant PFAS in the environment (Arp et al., 2024); presently, there are nearly 2,000 chemicals that have the potential of forming TFA in the environment (Adlunger et al., 2021). The atmospheric chemistry of HFO-1234ze(E) has been the subject of a few past studies

and is relatively well understood (Søndergaard et al., 2007; Javadi et al., 2008; Burkholder et al., 2015b). HFO-1234ze(E) primarily interacts with OH radicals in the atmosphere, leading to the formation of trifluoroacetic aldehyde (TFAA) as the main product (Javadi et al., 2008; Burkholder et al., 2015b). TFAA is mainly eliminated from the atmosphere through photolysis, a process that does not result in the formation of TFA (Tewari et al., 2023). However, it has been suggested that the OH-initiated TFAA atmospheric degradation and the subsequent reaction of the corresponding peroxy radicals with $HO_2$ radicals can lead to TFA formation (Sulbaek-Andersen et al., 2004). In addition, several other TFAA degradation pathways have been suggested in the literature (Pérez-Peña et al., 2023; Long et al., 2022; Sulbaek-Andersen and Nielsen, 2022) which, ultimately, can affect the overall atmospheric chemistry of HFO-1234ze(E) and therefore its potential to form TFA. To the best of our knowledge, except for a box-modelling study using a simplistic HFO-1234ze(E) degradation mechanism (Tewari et al., 2023), there is no other work that explored the formation of TFA from HFO-1234ze(E) utilizing a global chemical transport modelling. Lastly, provided that any TFA formed from the atmospheric degradation of fluorinated gases is, eventually, deposited on Earth's surface via wet and dry deposition, it is also important to evaluate its fate and transport in surface soil and water media, upon deposition.

We therefore performed a state-of-the-art study coupling global chemical transport modelling with TFA surface fate and transport modelling to estimate surface water, sediment, and surface soil concentrations of TFA over a period of 30 years due to continued, global sales of prospective pMDIs using only HFO-1234ze(E) as the medical propellant. We used IQVIA MIDAS® monthly volume sales for 2022, for 51 countries, to estimate the volume of propellant present in all pMDI units sold worldwide. Lastly, we compared our estimates of environmental TFA production due to pMDI usage with that from all other sources.

## 2 Methods

### 2.1 Software

We applied a global three-dimensional Eulerian chemical transport model, GEOS-Chem (version 14.2.2, https://doi.org/10.5281/zenodo.10034733), driven by meteorological data assimilated from the Goddard Earth Observing System (GEOS) at the National Aeronautics and Space Administration (NASA) Global Modeling and Assimilation Office. The model incorporates a detailed mechanism for oxidant-aerosol chemistry computed within the troposphere and the stratosphere at 30 minute time intervals using a fourth-order Rosenbrock kinetic solver implemented with the Kinetic preprocessor version 3.0 (Lin et al., 2023). Emissions were calculated at 30 minute time steps using the Harmonized Emissions Component (HEMCO) module version 3.0 (Lin et al., 2021).

In this study, GEOS-Chem was configured to use modern-era retrospective analysis for research and applications, version 2 (MERRA-2) reanalysis meteorology data with varying temporal resolutions: 3-hourly for three-dimensional fields, such as wind components (zonal wind and meridional wind) and temperature, and hourly for surface variables and mixing depths, including soil moisture, heat fluxes, and albedo (Gelaro et al., 2017). We conducted the global modeling at a 2°×2.5° resolution

across 47 vertical 'eta' levels from the surface to approximately 80 km; here, 'eta' refers to vertical levels in hybrid sigma-pressure coordinates. Additionally, the horizontal resolution of the meteorological fields was updated to align with the 2°×2.5° grid of the model. The wet deposition of aerosols and soluble gases by precipitation includes the scavenging in convective updrafts, in-cloud rainout, and below-cloud washout (Liu et al., 2001). The dry deposition was calculated using a resistance-in-series parameterization, which is dependent on environmental variables and lookup table values (Wesely, 2007). To obtain initial concentrations, model spin-up simulations were performed at 4°×5° grid resolution for approximately eight years until the concentrations of species varied little from one year to the next. The final simulations were performed at 2°×2.5° resolution for the year 2022.

## 2.2 Anthropogenic, Natural, and HFO-1234ze(E) emissions

The global anthropogenic emissions are from the Community Emissions Data System (CEDS version 2) inventory for 1980-2019 at 0.1°×0.1° resolution (Hoesly et al., 2018). They include chemically reactive gases (sulfur dioxide ($SO_2$), nitrogen oxides ($NO_x$), ammonia ($NH_3$), methane ($CH_4$), carbon monoxide (CO), and non-methane volatile organic compounds (NMVOCs)), carbonaceous aerosol (black carbon (BC) and organic carbon (OC)), and carbon dioxide ($CO_2$). Emissions are provided on an annual basis at the level of country and sector, with a monthly temporal resolution. The CEDS inventory includes emissions from ships for all species included in the inventory and is used as the global ship emissions inventory in the model. Ethane emissions were from Tzompa-Sosa et al. (2017) and propane emissions from Xiao et al. (2008). Aircraft emissions are from the AEIC 2019 inventory (Simone et al., 2013). The monthly-averaged 0.25°×0.25° biomass burning emissions were obtained from the Global Fire Emissions Database version 4 (GFED4, (Van Der Werf et al., 2017)). The biogenic volatile organic compounds emissions were derived from the Model of Emissions of Gases and Aerosols from Nature (MEGAN) version 2.1 (Guenther et al., 2012), as implemented by Hu et al. (2015), and calculated offline to improve reproducibility across scales (Weng et al., 2020). The mineral dust emissions were calculated offline at native meteorological resolution using the Dust Entrainment and Deposition scheme of Zender et al. (2003), combined with an updated high-resolution dust source function (Meng et al., 2020). Sea salt emissions from the open ocean are dependent on wind speed and sea surface temperature, and follow the algorithm of Jaegle et al. (2011). The algorithm for above-canopy soil $NO_x$ emissions follows Hudman et al. (2012), with the efficiency of loss to the canopy depending on vegetation type and density. Emissions from other natural sources (e.g., lightning and volcanoes) were also included (Carn et al., 2015; Murray et al., 2012).

To estimate global HFO-1234ze(E) emissions from prospective pMDI use, we used IQVIA MIDAS® monthly sales data for the top 51 countries in 2022 (see supplementary data), based on an assumed daily dose of 4 puffs per day and approximately 14 g of HFO-1234ze(E) per pMDI unit. These emissions were calculated as monthly totals and then spatially distributed within each country using NO emissions from the residential and commercial sectors in the CEDS inventory as a proxy. This method assumes that NO emissions in these sectors correlate with population density and, therefore, with likely pMDI usage. As a result, the spatial resolution of our emission estimates matches that of the CEDS proxy data and does not represent the true

distribution of pMDI use among patients with respiratory diseases. Global spatial patterns of estimated HFO-1234ze(E) emissions are discussed in Section 3.

## 2.3 HFO-1234ze(E) degradation mechanism

The standard GEOS-Chem model uses a set of chemical mechanisms implemented with a kinetic preprocessor. The model includes aerosol chemistry and stratospheric chemistry (Eastham et al., 2014), and reaction rates and products are based on NASA Jet Propulsion Laboratory (JPL) Panel for Data Evaluation or the International Union of Pure and Applied Chemistry (IUPAC) Task Group on Atmospheric Chemical Kinetic Data Evaluation recommendations (Burkholder et al., 2015a; Sander et al., 2006; Sander et al., 2010). Photolysis frequencies are calculated with the Fast-JX code (version 7.0) (Bian and Prather, 2002) as implemented in GEOS-Chem by Mao et al. (2010) for the troposphere and by Eastham et al. (2014) for the stratosphere. There are a total of 921 reactions, including 104 photolysis reactions, in the model.

A detailed gas phase degradation mechanism for HFO-1234ze(E) has been constructed, in a format that is compatible with that used in the Master Chemical Mechanism, MCM (Jenkin et al., 1997; Jenkin et al., 2003; Jenkin et al., 2015; Saunders et al., 2003). The mechanism makes use of published kinetic and mechanistic information relevant to HFO-1234ze(E) degradation and, where possible, applies parameters recommended either by the IUPAC https://iupac.aeris-data.frhttp://jpldataeval.jpl.nasa.gov/ or the JPL. Where no information is available, kinetic parameters and product channel contributions are estimated using published estimation methods, e.g., Jenkin et al. (2019), Kwok and Atkinson (1995), or assigned by analogy with chemistry reported for structurally similar species. Figure 1 shows the full mechanism, which contains 68 reactions. Table S1 of the Supplement describes these reactions and the origin of the assigned kinetic parameters. To reduce the computational burden of the GEOS-Chem model, we have simplified the mechanism to twenty-nine reactions, including four photolysis reactions and reactions associated with TFA formation. We have excluded some atmospheric species, such as $O_2$ and $H_2O$, from the chemical reactions since concentrations of these species are provided by the meteorological input data (MERRA2) and including them explicitly in every chemical reaction can add unnecessary complexity without significantly improving the model accuracy.

The first step is the hydroxyl radical (OH)-initiated oxidation of $CF_3CH=CHF$, represented by a single reaction in GEOS-Chem (Tewari et al., 2023):

$$CF_3CHCHF + OH\ (+O_2) \rightarrow FRO_2 \hspace{4cm} (R1)$$

Here, $FRO_2$ denotes the peroxy-radicals $CF_3CH(OH)CH(O_2)F$ and $CF_3CH(O_2)CH(OH)F$. The rate constant for reaction R1 is from the IUPAC (Ammann et al., 2016). The peroxy radical from reaction R1 reacts with NO and $HO_2$. The reaction with NO generates 2,2,2- trifluoroacetic aldehyde (TFAA; $CF_3CHO$) and formyl fluoride (HCOF), while its reaction with $HO_2$ forms a hydroxy-carbonyl product, $CF_3CH(OH)C(O)F$:

$$FRO_2 + NO \rightarrow CF_3CHO + HCOF + HO_2 + NO_2 \hspace{3cm} (R2)$$

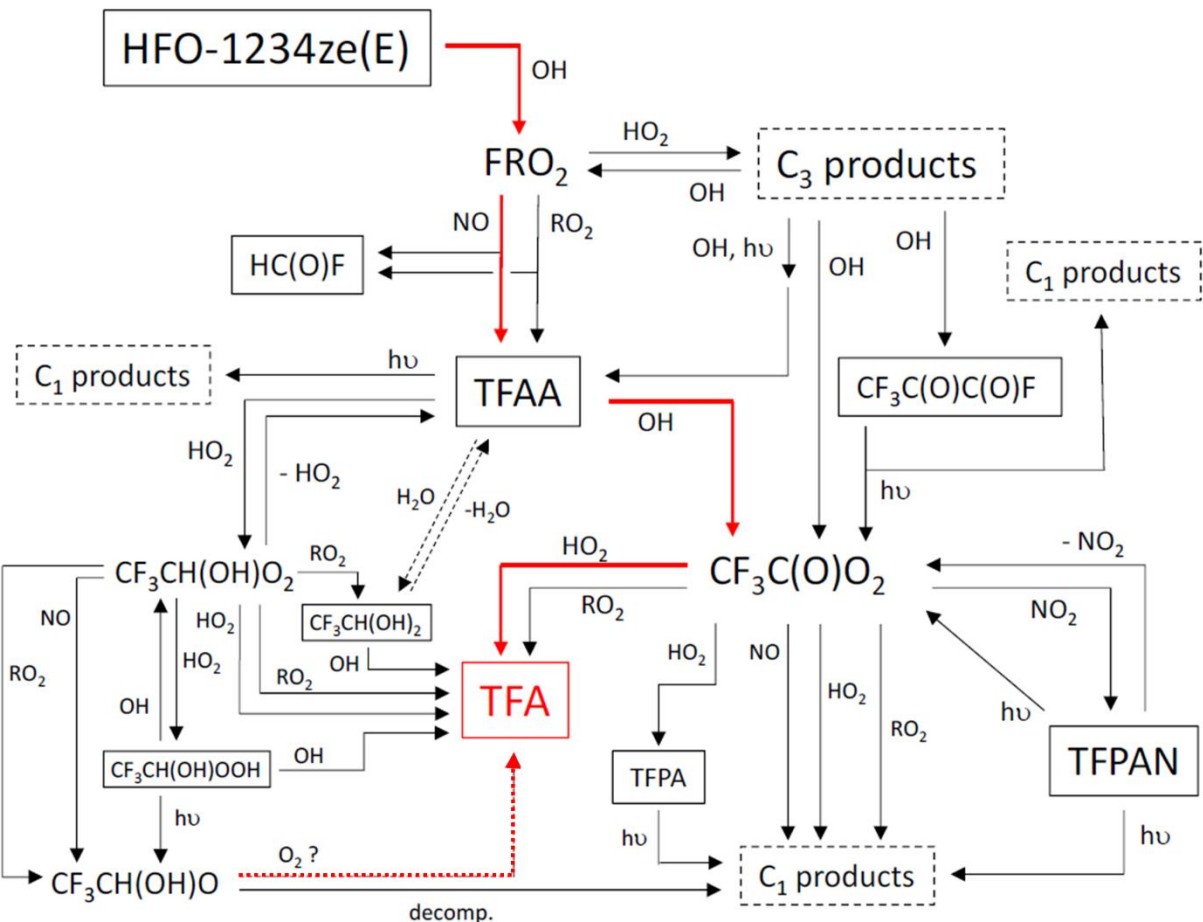

**Figure 1:** Schematic of gas-phase degradation mechanism for HFO-1234ze(E). Solid red arrows show the primary route for TFA (trifluoroacetic acid, $CF_3C(O)OH$) formation; dashed red arrow shows the theoretical route of TFA formation via the reaction of TFAA with $HO_2$ (Long et al., 2022). The in-cloud hydration of TFAA (shown using dotted lines) followed by its outgassing and reaction with OH radicals has been discussed as a possible route to TFA formation; however, due to lack of sufficient information, this pathway has not been considered in the existing mechanism. Abbreviations: $FRO_2$, peroxy radicals $CF_3CH(OH)CH(O_2)F$ and $CF_3CH(O_2)CH(OH)F$; TFAA, trifluoroacetic aldehyde; TFPAN, trifluoroacetyl peroxy nitrate; TFPA, trifluoroperacetic acid; and C3 products, $CF_3CH(OH)CH(OOH)F$, $CF_3CH(OOH)CH(OH)F$, $CF_3CH(OH)C(O)F$ and $CF_3C(O)CH(OH)F$.

$$FRO_2 + HO_2 \rightarrow CF_3CH(OH)C(O)F + H_2O + O_2 \qquad \text{(R3)}$$

The rate constants for reactions R2 and R3 are based on IUPAC recommendations for similar species (Ammann et al., 2016).

The hydroxy-carbonyl product (R3) is removed from the atmosphere by reacting with hydroxyl radical forming 3,3,3-trifluoro-

2-oxo-propionyl fluoride, $CF_3C(O)C(O)F$, and $CF_3CHO$:

$$CF_3CH(OH)C(O)F + OH\ (+O_2) \rightarrow CF_3C(O)C(O)F + HO_2 \qquad \text{(R4a)}$$

$$CF_3CH(OH)C(O)F + OH\ (+O_2) \rightarrow CF_3CHO + FCO_3 \qquad \text{(R4b)}$$

We included the photolysis of $CF_3C(O)C(O)F$, which forms acyl peroxy radicals ($CF_3C(O)O_2$) given as:

$$CF_3C(O)C(O)F + h\nu \rightarrow CF_3C(O)O_2 \tag{R4c}$$

A wavelength dependent cross-section of $CH_3C(O)CHO$ (Ammann et al., 2016) was used for $CF_3C(O)C(O)F$ as shown in Fig. S1 of the Supplement. The wavelength dependent quantum yield is also based on $CH_3C(O)CHO$ and is 1 between 225-380 nm but decreases at higher wavelengths.

There are two major pathways for the degradation of $CF_3CHO$ in the atmosphere. First, it undergoes photolysis. The initial products formed via photolysis undergo a series of additional reactions to give $CO_2$ and hydrogen fluoride (HF) (Sulbaek-Andersen et al., 2018):

$$CF_3CHO + h\nu \rightarrow CO_2 + HF \tag{R5a}$$

For reaction R5a, we applied a wavelength-dependent cross-section (Ammann et al., 2016), shown in Fig. S1 in the Supplement. A wavelength-dependent quantum yield was calculated considering the recommended values from IUPAC (Ammann et al., 2016) and values from Sulbaek-Andersen and Nielsen (2022). Second, oxidation initiated by OH radical produces $CF_3C(O)O_2$, which can react with $HO_2$, NO, and $NO_2$.

$$CF_3CHO + OH\ (+O_2) \rightarrow CF_3C(O)O_2 + H_2O \tag{R5b}$$

Because IUPAC only provides a rate recommendation for R5b at 298 K (Ammann et al., 2016), we determined the temperature dependence of R5b using IUPAC recommendations for $CCl_3CHO$, which has comparable reactivity. There are several reactions and reaction channels for $CF_3C(O)O_2$ that compete with its TFA-forming reaction with $HO_2$:

$$CF_3C(O)O_2 + NO \rightarrow CF_3C(O)O + NO_2 \tag{R6a}$$

$$CF_3C(O)O_2 + NO_2\ (+M) \rightarrow CF_3C(O)OONO_2\ (+M) \tag{R6b}$$

$$CF_3C(O)OONO_2\ (+M) \rightarrow CF_3C(O)O_2 + NO_2\ (+M) \tag{R6c}$$

$$CF_3C(O)O_2 + HO_2 \rightarrow CF_3C(O)OOH + O_2 \tag{R7a}$$

$$CF_3C(O)O_2 + HO_2 \rightarrow CF_3C(O)OH + O_3 \tag{R7b}$$

$$CF_3C(O)O_2 + HO_2 \rightarrow CF_3C(O)O + O_2 + OH \tag{R7c}$$

According to kinetic analysis data from Maricq et al. (1996) and Wallington et al. (1994), IUPAC recommends a rate constant for reaction R6a of $4.02\times10^{-12}\exp(560/T)$ $cm^3$ molecule$^{-1}$ s$^{-1}$. Reaction R6b is a termolecular reaction, with its rate coefficient depending on pressure. The IUPAC recommended value for this rate coefficient is based on data from Wallington et al. (1994). The product of reaction R6b is trifluoroacetyl peroxy nitrate (TFPAN), which is thermally unstable and dissociates back into reactants (reaction R6c) at higher temperatures, exhibiting strong temperature dependence. The TFPAN lifetime ranges from approximately 0.5 days at the Earth's surface to several months or even years at higher altitudes (Ammann et al., 2016).

Consequently, TFPAN is expected to undergo significant transport within the troposphere and can serve as a reservoir for $CF_3C(O)O_2$, which may be released again in warmer regions. In addition to R6c, TFPAN can also photolyze:

$$CF_3C(O)OONO_2 + h\nu \rightarrow 0.5CF_3C(O)O_2 + 0.5NO_2 + 0.5CF_3C(O)O + 0.5NO_3$$

The assigned photolysis rates for TFPAN are based on the absorption cross sections recommended by the NASA JPL (Burkholder et al., 2020), and typically result in a lifetime with respect to photolysis of about 2 or 3 weeks, with photolysis becoming the major loss reaction at altitudes above about 4 km. Reactions R7a-R7c comprise a complex sequence of processes, with reaction R7b forming TFA ($CF_3C(O)OH$). Because the branching ratios for R7a-c have been evaluated at only 296 K by (Sulbaek-Andersen et al., 2004), we have considered them to be temperature independent in this study. Among these reactions,
R7c is predominant with a branching ratio of $0.56 \pm 0.05$, followed by R7b with $0.38 \pm 0.04$ and R7a with $0.09 \pm 0.04$. By analogy with the corresponding reaction for $CH_3C(O)O_2$, the contribution of channel (R7a) might be expected to increase at lower temperatures and needs to be experimentally investigated. We have also considered the reaction of $CF_3C(O)O_2$ with the tropospheric pool of peroxy radicals, $RO_2$, i.e., $CH_3O_2$. The reaction in this case was based on the IUPAC recommendation for the reaction of $CH_3C(O)O_2$ with $CH_3O_2$:

$$CF_3C(O)O_2 + CH_3O_2 \rightarrow CF_3C(O)O \qquad \text{(R8a)}$$

$$CF_3C(O)O_2 + CH_3O_2 \rightarrow CF_3C(O)OH \qquad \text{(R8b)}$$

Reaction R8b provides an additional pathway to TFA formation but is calculated to be an order of magnitude less important than reaction R7b. $CF_3CHO$ formed in reaction R4b can react with $HO_2$ (reaction R5c) that has been characterized in a theoretical study by Long et al. (2022).

$$CF_3CHO + HO_2 \rightarrow CF_3CH(OH)O_2 \qquad \text{(R5c)}$$

Under most tropospheric conditions, the significance of reaction R5c is constrained by the rapid thermal decomposition of the peroxy radical, $CF_3CH(OH)O_2$, which produces $CF_3CHO$ and $HO_2$ (reaction R9a), as reported by Long et al. (2022). However, this decomposition rate is highly temperature dependent. At the lower temperatures found in the upper troposphere, the subsequent reactions of $CF_3CH(OH)O_2$ with NO (reaction R9b) and $HO_2$ (reaction R9c) become competitive, potentially
creating pathways for TFA formation. Reaction R9b combines two potential pathways for the initially formed intermediate species $CF_3CH(OH)O$, which can decompose to form HCOOH (Orlando et al., 2000; Jenkin et al., 2005) and $CF_3C(O)O$ or can react with $O_2$ to form TFA and $HO_2$. While the decomposition pathway is expected to be significant throughout the troposphere, reaction with $O_2$ may be competitive at high altitudes and low temperatures and its inclusion in the chemical mechanism likely provides an upper estimate of TFA formation.

$$CF_3CH(OH)O_2 \rightarrow CF_3CHO + HO_2 \qquad \text{(R9a)}$$

$$CF_3CH(OH)O_2 + NO \rightarrow 0.5HCOOH + 0.5CF_3C(O)O + 0.5CF_3C(O)OH + 0.5HO_2 + NO_2 \qquad \text{(R9b)}$$

$$CF_3CH(OH)O_2 + HO_2 \rightarrow 0.5CF_3CH(OH)OOH + 0.5CF_3C(O)OH + 0.2OH + 0.2HO_2 \qquad \text{(R9c)}$$

The impact of reactions R9a, R9b, and R9c on TFA formation is critically dependent on the competition between the thermal decomposition of the peroxy radical (R9a) and its reaction with NO (R9b) and HO$_2$ (R9c). Additionally, CF$_3$CH(OH)OOH can be oxidized by the OH radical and can also undergo photolysis, and reactions R10a and R10c have the potential to lead to TFA formation.

$$CF_3CH(OH)OOH + OH \rightarrow CF_3C(O)OH + OH + H_2O \qquad \text{(R10a)}$$

$$CF_3CH(OH)OOH + OH \rightarrow CF_3CH(OH)O_2 \ + H_2O \qquad \text{(R10b)}$$

$$CF_3CH(OH)OOH + h\nu \rightarrow 0.5CF_3C(O)O + 0.5HCOOH + 0.5CF_3C(O)OH + 0.5HO_2 + OH \qquad \text{(R10c)}$$

A wavelength dependent cross-section for CH$_3$OOH was applied to CF$_3$CH(OH)OOH in reaction R10c as recommended by IUPAC (Fig. S1 in the Supplement). Gas-phase TFA is expected to be deposited either through dry or wet deposition or by reacting with OH radicals as follows:

$$CF_3C(O)OH + OH\ (+O_2) \rightarrow CF_3C(O)O + CO_2 \qquad \text{(R11)}$$

**2.4 Surface fate and transport modelling of TFA**

We next performed modeling to assess the fate and transport of TFA in the surface water resulting from environmental TFA deposition due to atmospheric degradation of HFO-1234ze(E) via reactions presented in Section 2.3. To this end, we chose the Hudson River, Rhine River, and the Cauvery River as they represent prominent watersheds, also known as drainage basin or catchment area,  with relatively large populations (see Figs. S2-S4 in the Supplement). In this study, we chose the Rhine watershed in the Europe as the primary location due to the availability of numerous TFA-related literature references for comparison. The Hudson (USA) watershed was selected because *i*) the USA has the highest pMDI volume sales globally and Hudson has proximity to New York City, which has the country's highest population density and, thus, presumably, has the highest pMDI usage, i.e., propellant emissions, with the USA.

Note even though the phasedown of HFCs in India will take longer than in the USA or Europe, we chose India as the third watershed because it has the third-largest pMDI volume sales in the world, after the USA and the UK. Since we were already considering the Rhine watershed (Europe), we decided to include India instead of the UK as the third location. The selection of the Cauvery watershed within India was guided by the availability of watershed modelling parameters and the fact that it represents a source of drinking water to a very large population centre, i.e., Bengaluru (see Fig. S4).

The TFA concentrations in the surface water, sediment, and soil in the Hudson River, Rhine River, and Cauvery River watersheds were simulated through fate and transport modeling based on the model-predicted annual average deposition rates (see Results & Discussion). The modeling methodology aligns with the USEPA Human Health Risk Assessment Protocol

(HHRAP) for Hazardous Waste Combustion Facilities (Epa, 2005). We determined the TFA loading of the water column using the following mechanisms: *1)* Direct deposition, *2)* Runoff from surfaces within the watershed, *3)* Soil erosion over the total watershed, *4)* Benthic burial, *5)* Inputs from the upstream river segment, and *6)* Discharge to ocean. We assumed that the contributions from other potential mechanisms as compared to those listed above are negligible. Because TFA is resistant to degradation in the environment, we do not account for chemical or biological transformation of TFA after it is deposited on the ground or on surface water. In this model, we have not considered evaporative loss of TFA into the environment because TFA ionizes in water and should not evaporate in the river. Table 1 lists the physical and chemical properties of TFA used in the modeling.

**Table 1. Physical/Chemical Properties for TFA**

| Parameter | Value |
|---|---|
| Organic Carbon-Water Partition Coefficient[1] | 6.22 L kg$^{-1}$ |
| Soil-Water Partition Coefficient[2] | 0.94 L kg$^{-1}$ |
| Soil Enrichment Ratio[3] | 3 |
| Suspended Sediment/Surface Water Partition Coefficient[4] | 0.47 L kg$^{-1}$ |
| Bed Sediment/Sediment Pore Water Partition Coefficient[5] | 0.25 L kg$^{-1}$ |
| Diffusivity in Water* | $8.00 \times 10^{-6}$ cm$^2$ s$^{-1}$ |

[1] calculated according to the equation from Sabljic and Güsten (Güsten and Sabljic, 1995)), using the class of non-hydrophobic chemicals. In the case of TFA, the class "organic acid" is more relevant. Therefore the Koc is calculated as follows: logKoc = 0.6 * logKow + 0.32, with LogKow = 0.79 (ECHA TFA Endpoint Summary for Transport and Distribution - Adsorption/Desorption: https://echa.europa.eu/registration-dossier/-/registered-dossier/5203/5/5/2). [2] Adsorption/desorption tests results show that TFA is poorly adsorbed to the soil and is considered as a mobile organic compound in the majority of soils investigated. The Kd ranged between 0.17 to 20 L/kg for organic and mineral soils (the organic horizon exhibiting greater retention) giving a geometric mean of 0.94 L/kg (SD=4.86, n= 20) (ECHA TFA Endpoint Summary for Transport and Distribution - Adsorption/Desorption: https://echa.europa.eu/registration-dossier/-/registered-dossier/5203/5/5/2). [3] Default value for organic compounds ((Epa, 2005), Appendix B, Table B-4-11)
[4] Calculated using the Koc and a default mid-range value of surface water foc of 0.075 ((Epa, 2005), Appendix A-2). [5] Calculated using the Koc and a default mid-range value of sediment foc of 0.04 ((Epa, 2005), Appendix A-2). * Taken from George et al. (1994). Notes: cm$^2$ s$^{-1}$: Square centimetre per second; foc: Fraction of organic carbon; L kg$^{-1}$: Liter per kilogram; Koc: Organic Carbon-Water Partition Coefficient

To simulate mobilization of TFA in watershed sub-basins, we used a series of compartment models that represent the sub-basins in a river system from the headwaters to the mouth of the river where it discharges into the ocean (Figure 2). Note that for Hudson watershed, we did not evaluate the lower Hudson River sub-basin because this section of the Hudson is generally the zone of greatest mixing of river water and the Atlantic Ocean.

Our surface fate and transport model predicts the steady-state concentrations of TFA in the water column and sediment layer beneath the water column. It does not account for the fluctuating flow of TFA between the water column and sediment in response to variations in external inputs. The overall concentration of TFA is divided between the sediment and the water column. Per HHRAP Guidance, the rate of soil erosion from the watershed is calculated by the Universal Soil Loss Equation (USLE) and a sediment delivery ratio. The sum of the TFA concentration dissolved in water and the TFA concentration

**Hudson River watershed**

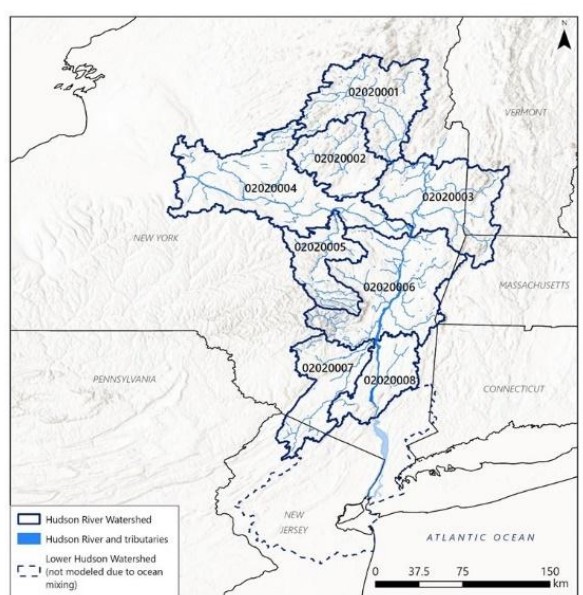

**Cauvery River watershed**

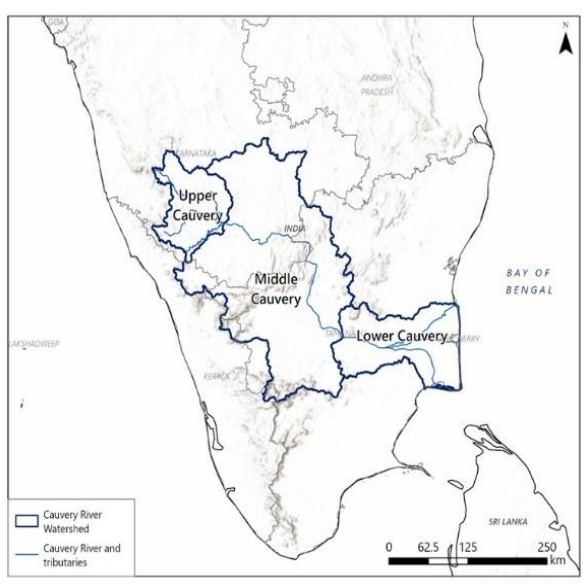

**Rhine River watershed**

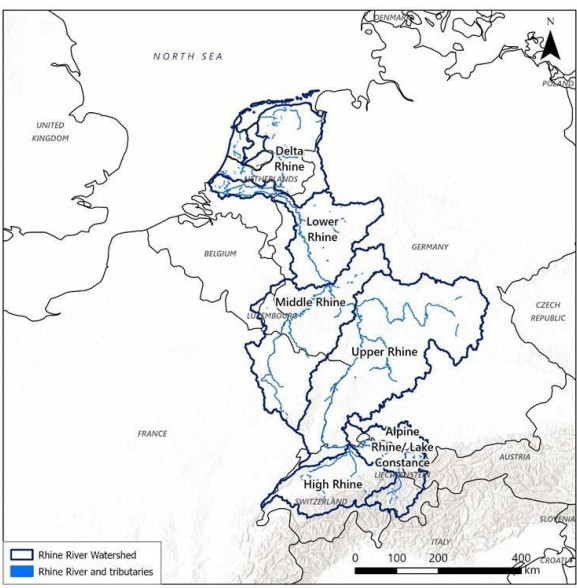

**Figure 2**. Sub-Basins in the Hudson River watershed (upper-left), Cauvery River watershed (upper-right), and Rhine River watershed (bottom). Note that we modelled the Hudson River watershed assuming the following three sub-basins: Sub-basin #1: 02020001 and 02020002; Sub-basin #2: 02020003, 02020004 and 02020005; and Sub-basin #3: 02020006, 02020007 and 02020008. The lower Hudson River sub-watershed was not evaluated because this zone has high mixing of river water and the Atlantic Ocean.

associated with suspended solids constitutes the total estimated water column TFA concentration. The detailed methodology and governing equations used in the modelling are outlined in the USEPA HHRAP Guidance (Epa, 2005). In the Supplement

(Tables S2-S7), we have summarized the watershed and waterbody parameters for each sub-basin for the Hudson River, Rhine River, and Cauvery River. Some default modelling parameters are available, and these recommended values typically reflect national average conditions in the United States, such as the default empirical intercept coefficient of 0.6 for watersheds larger than 1000 square miles or 2560 square kilometres from the USEPA HHRAP (Epa, 2005) which was used in the modelling for all three rivers. The uncertainties associated with the selection of the watershed and waterbody parameters are also discussed in Section S1 of the Supplement.

## 3. Results and Discussion

### 3.1 Global HFO-1234ze(E) emissions and TFA deposition flux

We used monthly pMDI volume sales data (IQVIA MIDAS®) for the period 2022 from top 51 countries to calculate the total NGP emissions per month per country, assuming daily dose of 4 puffs per day and ~14 g of HFO-1234ze(E) per pMDI (see supplemental data). The spatial distribution of HFO-1234ze(E) emissions within each country in the GEOS-Chem model was estimated using anthropogenic nitric oxide

| Table 2. Estimated HFO-1234ze(E) emission flux | |
|---|---|
| Regions | HFO-1234ze(E) emissions |
| United States | $2.38 \times 10^{-5}$ kg m$^{-2}$ yr$^{-1}$ |
| Europe | $4.26 \times 10^{-5}$ kg m$^{-2}$ yr$^{-1}$ |
| Asia | $1.58 \times 10^{-5}$ kg m$^{-2}$ yr$^{-1}$ |
| South America | $0.799 \times 10^{-5}$ kg m$^{-2}$ yr$^{-1}$ |
| Rest of the world | $1.21 \times 10^{-5}$ kg m$^{-2}$ yr$^{-1}$ |
| Global emissions flux | $4.736 \times 10^{6}$ kg yr$^{-1}$ |

(NO) emissions from the CEDS inventory as a proxy, based on the assumption that NO emissions from residential and commercial sectors correlate with population density and, hence, pMDI usage. Figure 3 shows the annual spatial distribution of HFO-1234ze(E) emissions at a 2°×2.5° resolution in GEOS-Chem, and Table 2 provides the HFO-1234ze(E) emissions in different regions of the world and the entire world. The worldwide emissions of HFO-1234ze(E) from pMDI use in the 51 countries are estimated to be 4.736 Gg/yr (or kilotons/year). Although respiratory disease symptoms can vary by season, our estimates of global HFO-1234ze(E) emissions show small variations with different seasons suggesting sustained demand of pMDIs in the respiratory disease community (Fig. S5 in the Supplement). This apparent consistency is primarily because different regions of the world experience their peak NGP emissions at different times of the year. For example, if you compare the monthly emissions data for the UK and Brazil in the Supplement, you will see different peak periods. When these regional trends are averaged at the global level, these seasonal differences are effectively smoothed out.

We followed the formulation of Wesely (2007), subsequently modified by Wang et al. (1998), for computation of dry deposition of gas-phase species onto surfaces in GEOS-Chem, which calculates dry deposition velocities using data for surface momentum, sensible heat fluxes, temperature, and

solar radiation. To incorporate the dry deposition of TFA, we assumed its behaviour in this process to be analogous to that of nitric acid, following the approach of Luecken et al. (2010) and Henne et al. (2012). Figure 4 shows the modelled annual global dry and wet deposition fluxes of TFA. While previous

studies (for example, Luecken et al., 2010; Henne et al., 2012; Wang et al., 2018; Sulbaek Andersen et al., 2018) have concluded that wet deposition is the dominant pathway for TFA removal, our results indicate that dry deposition may

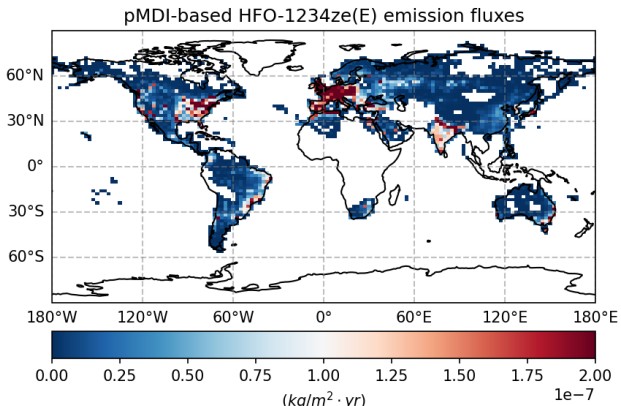

**Figure 3:** Global annual distribution of HFO-1234ze(E) emissions at 2°×2.5° resolution in GEOS-Chem. This distribution of HFO-1234ze(E) emissions is based on global pMDI volume sales data, assuming pMDI devices sold by all manufacturers in 2022 used HFO-1234ze(E) as the propellant.

play a more significant role under certain tropical and subtropical conditions. This divergence is likely due to differences in

the chemical formation of TFA—particularly via TFAA—and in regional meteorological patterns that influence deposition processes.

The monthly variations in global dry and wet deposition of TFA are shown in Figs. S6–S7 in the Supplement, which capture the effect of different seasons on TFA deposition fluxes globally. We note that TFA tends to have higher deposition rates in

the summer as refrigerants are more commonly used in warmer months and more oxidants are present in summer, although precipitation is also a factor as it allows for more wet deposition. Wang et al. (2018) also noted that maximum wet and dry

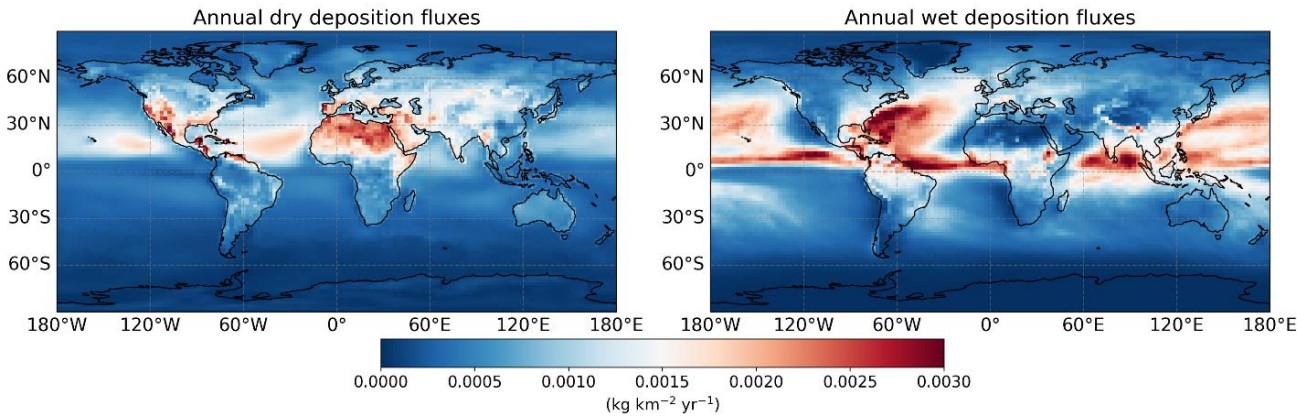

**Figure 4:** Annual global dry and wet deposition fluxes of TFA. These fluxes reflect the average of TFA deposited over a year. Note that dry deposition fluxes are highest in African region and wet deposition fluxes are highest around the ocean near the northern tropical region

TFA deposition occurred during the summer, accounting for 71% of global yearly deposition, followed by autumn with 12% global yearly deposition. Wang et al. (2018) also stated that although parent-molecule emissions are the dominant factor for TFA deposition, but convective mass flux and precipitation also have an effect.

We found that the total deposition between 0° and 45° north latitudes is approximately 61% of the global total deposition. Even though NGP emission fluxes are four times lower in Africa as compared to Europe (Table 2), the dry deposition fluxes are higher in Africa as compared to Europe. This is presumably due to the higher concentration of hydroxyl radicals around the equator (Pimlott et al., 2022) that will increase the formation of $CF_3CO(O)_2$ from TFAA via reaction 5b (R5b). However,

transport of other intermediate species, such as TFA itself, into the tropical region as a cause of higher TFA deposition flux cannot be completely ruled out.

To understand the effects of seasonal variation and altitude on TFA-deposition fluxes in two identified regions in Europe and Africa (Fig. S8 in the Supplement), we computed correlation between daily TFAA and TFA formed per day for each month at

different altitudes (Fig. S9 in the Supplement). We found a negative correlation between TFAA and TFA in the identified regions of Europe and Africa for almost the entire year, confirming that the hydroxyl radical channel (R5b) is the primary route of TFA formation under different environmental conditions. In contrast to this general agreement, we found that reaction channels other than R5b contribute to TFA formation during May thru September in the identified region of Africa (Fig. S9 in the Supplement; Spearman's $\rho \approx 0$).

To understand contribution of primary TFA-forming pathways on the spatial pattern of TFA, we examined concentrations of various-related species at the surface after the last spin-up simulations at 4°×5° resolution (see Methods). Our assessment of initial oxidation products suggests that HFO-1234ze(E) emissions at the surface largely correlate with corresponding TFAA concentrations at the surface (Section S2; Fig. S10d) and show little correspondence with the initial oxidation products (Fig.

S10b,c). The subsequent degradation of TFAA can occur via OH (reaction R5b) or $HO_2$ (reaction R5c); of these two possible routes, the OH-pathway which forms CF3C(O)O$_2$ is about three orders of magnitude faster as compared to the $HO_2$ pathway (Table S1), leading to more $CF_3C(O)O_2$ concentrations (R5b) than $CF_3CH(OH)O_2$ (R5c), see Fig. S11 in the Supplement. Lastly, we calculated the ratio $[HO_2] \times [CF_3C(O)O_2]/[OH]$ to quantify the contribution of species that form TFA (R7b) and that remove TFA from the atmosphere (R11). Figure 5 compares the gas-phase concentrations of TFA and

$[HO_2] \times [CF_3C(O)O_2]/[OH]$ at the surface. This analysis suggests that *1*) reactions that form TFA generally dominate over those that remove it in the atmosphere, and *2*) $CF_3C(O)O_2$ reaction is the primary precursor that forms TFA in the atmosphere.

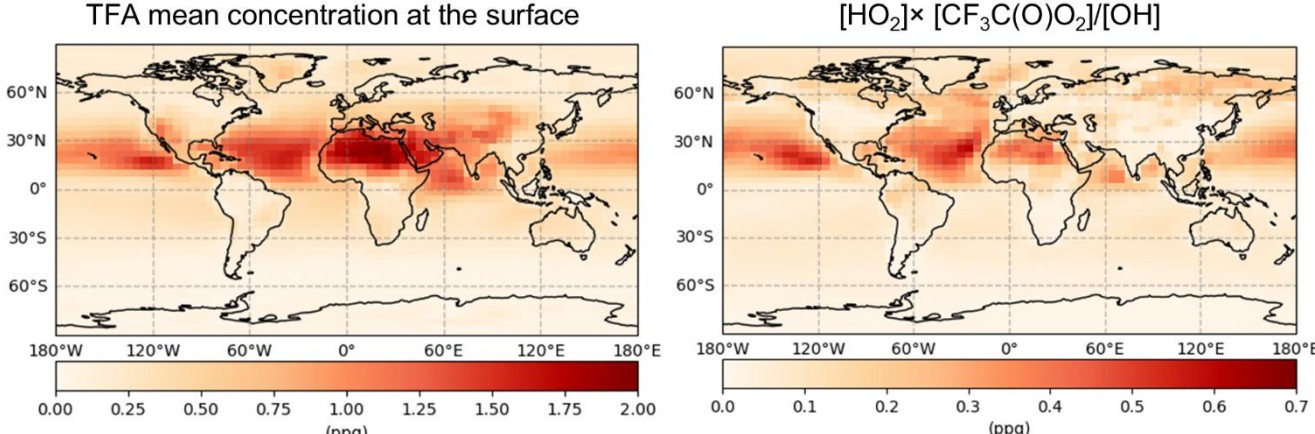

**Figure 5**: Annual mean-surface concentrations (gas-phase) of TFA and ([HO$_2$] × [CF$_3$C(O)O$_2$])/[OH] at 4°×5° resolution. The spatial pattern of the ratio of ([HO$_2$] x [CF$_3$C(O)O$_2$])/[OH] generally matches the spatial pattern of the TFA.

In general, the steady-state concentration of any atmospheric species is governed by processes that control both its production and removal; in the case of gas-phase TFA, however, our analysis reveals a distinct scenario. The key detail is that gas-phase
TFA is removed from the atmosphere almost exclusively through a single OH-driven, temperature-independent oxidation process (R11; Reaction 67 in Table S1). The rate of this removal pathway is significantly lower—by about an order of magnitude—than the rates of the major TFA formation reactions, including the OH-dependent reaction that forms TFA. As a result, the atmospheric burden and spatial patterns of gas-phase TFA are primarily determined by spatial patterns of chemical species that form TFA, rather than the ones that remove it.

We demonstrate this by showing that the spatial distribution of TFA closely matches that of the ratio [HO$_2$]×[CF$_3$C(O)O$_2$]/[OH], which includes species that form and remove TFA in the atmosphere. The numerator reflects the major formation processes, while the denominator includes the OH species responsible for both TFA formation and removal. Thus, the observed spatial correlation in the figure is not simply a reflection of two unrelated quantities but encapsulates the dominant chemistry controlling TFA's presence in the atmosphere. Section S2 of the supplement provides the entire process
which shows atmospheric concentration of other crucial intermediates. This section demonstrates that the spatial patterns observed for TFA are distinct and are not mirrored by other intermediate species in the same way, supporting our mechanistic interpretation.

### 3.2 Fate and transport modelling of pMDI-derived TFA

This section presents and discusses the modelled TFA concentrations in the surface water, surface sediment, and surface soil
for the sub-basins of each of the three river watersheds and the mass balance of TFA in these systems. We have used the methodology described in Section 2.4 for estimating the TFA concentrations in the sub-basins of the Hudson, Rhine, and Cauvery River watersheds. The model is based on a series of connected compartment models for each sub-basin of the river

system. The input of TFA from atmospheric deposition into the compartment for a sub-basin is based on area-weighted average (calculated based on the % of the grid cell areas overlapping with the sub-basin) TFA deposition flux for that sub-basin. Figure 6 shows the grid cells encasing spatial distribution of the total (dry + wet) TFA deposition flux and Table 3 lists weighted-average TFA deposition flux for each sub-basin of the three watersheds. Note that the distinct colour scales in Figure 6 capture differences in the total TFA deposition flux values in the three watersheds.

Assuming no inter-annual variability in the annual TFA deposition fluxes (Table 3), we estimated net TFA accumulations in the surface water, soil, and sediment of the three watersheds over a 30-year period. While TFA deposition varies seasonally, the focus here is on the accumulation of TFA over the 30-year period. Also, it is unknown how the seasonality itself would vary over the 30-year period. So the surface

| Table 3. Annual Average Atmospheric Deposition Flux of TFA ($\mu g\ m^{-2}\ yr^{-1}$) | | | |
|---|---|---|---|
| Sub-Basin | Dry Deposition | Wet Deposition | Total Deposition |
| **Hudson River watershed** | | | |
| 1 | 1.314 | 1.159 | 2.472 |
| 2 | 1.371 | 1.237 | 2.608 |
| 3 | 1.430 | 1.321 | 2.751 |
| **Cauvery River watershed** | | | |
| 1 | 1.520 | 2.618 | 4.138 |
| 2 | 1.637 | 1.879 | 3.515 |
| 3 | 1.414 | 1.857 | 3.271 |
| **Rhine River watershed** | | | |
| 1 | 1.207 | 1.539 | 2.746 |
| 2 | 1.308 | 1.618 | 2.926 |
| 3 | 1.354 | 1.104 | 2.459 |
| 4 | 1.293 | 1.035 | 2.328 |
| 5 | 1.152 | 0.967 | 2.119 |
| 6 | 0.956 | 1.002 | 1.958 |

fate and transport modeling was based on the mean annual deposition flux. Based on the simulation results, the TFA concentrations in various river systems reach a steady state at different rates (Figure 7), influenced by factors such as volumetric flow rates and deposition rates across sub- basins. We found that TFA concentrations reached steady state within one year in the Hudson River. Model predicted peak surface water TFA concentrations range from 2 ng L$^{-1}$ in the upstream sub-basin (sub-basin 1) to 5.4 ng L$^{-1}$ in the middle sub-basin (sub-basin 2). We found that these TFA accumulation variations

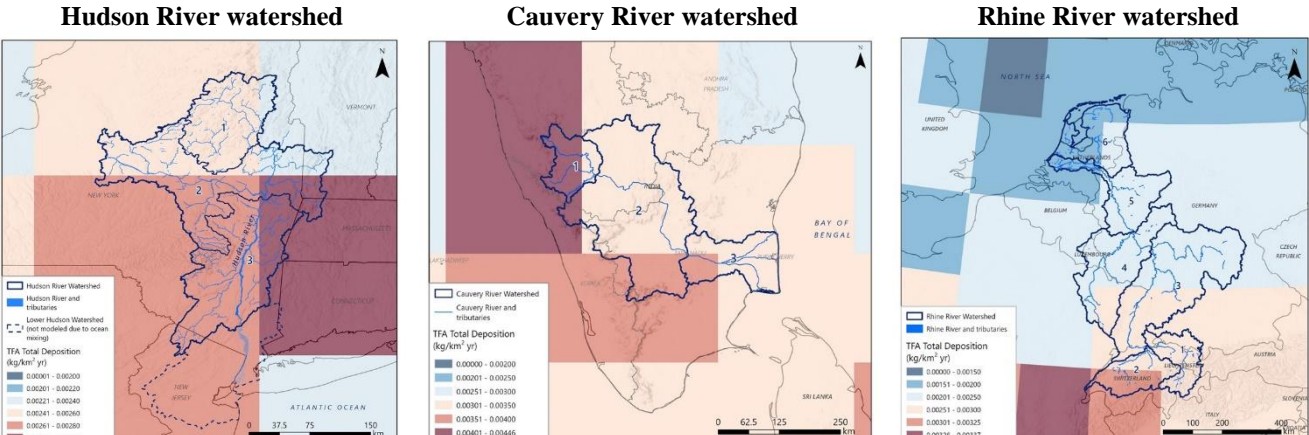

**Figure 6**. Estimated total TFA gridded deposition in the Hudson River watershed and surrounding area, in the Cauvery River watershed and surrounding area, and in the Rhine River watershed and surrounding area. Note that the color scale changes for each figure to display the range of deposition values within the extent of the map. *Cauvery watershed*: Sub-basin 1, Upper Cauvery; Sub-basin 2, Middle Cauvery; Sub-basin 3, Lower Cauvery. *Rhine watershed*: Sub-basin 1, Alpine Rhine; 2, High Rhine; 3, Upper Rhine; 4, Middle Rhine; 5, Lower Rhine; and 6, Delta Rhine. Hudson sub-basins are as described in the Method section.

within the sub-basins are impacted by factors such as differences in River flow rates, infiltration rates of the watersheds (see
Tables S2-S7 in the Supplement), and deposition rates (Table 3). Based on the simulation results of the TFA mass allocation
in soil, sediment, surface water in the river, and the ocean, approximately 63% of the TFA mass deposited from the air would
be delivered to the ocean.

Akin to the Hudson River results, we found that peak TFA surface water concentrations in Cauvery sub-basins also reached
steady state within a year. The model-predicted TFA concentrations range from 8.9 ng $L^{-1}$ in the upper Cauvery sub-basin to
19.3 ng $L^{-1}$ in the middle Cauvery sub-basin. Based on our TFA mass-allocation calculations, roughly 19% of the TFA mass
deposited from the air would be transported to the ocean, a small percentage (~4%) would remain in the river system (i.e.,
surface water and surface sediment) or the mixing zone of surface soil receiving deposition (i.e., assumed to be the top 2
centimeters of soil for this study), and roughly four-fifths would leach to deep soil or sediment, after the TFA concentrations
reach steady state in the system (Figure 7). The TFA mass allocation percentage to deep soil is higher for the Cauvery River
basin than that for Hudson River basin and Rhine River basin (next paragraph) because Cauvery River is located in the
tropical/sub-tropical zone and has much higher precipitation rates, and therefore, higher infiltration rates (determined by
precipitation, irrigation, run-off, and evapotranspiration in the basin) which caused increased leaching process of TFA to deep
soil in this basin. Note that TFA is highly mobile in soil due to its strong tendency to remain dissolved in water rather than
binding to soil particles, so it does not permanently accumulate in soil but quickly leaches into deeper layers and potentially
groundwater. Therefore, by accumulation in soil herein, we are referring to temporary retention of TFA in soil before leaching
and does not indicate long-term storage of TFA in soil.

Akin to the other two watersheds, the peak surface water concentrations of TFA in the Rhine River sub-basins also reached
steady state within a year, suggesting that region-specific TFA deposition differences are unsubstantial compared to River-
specific flow rates' impact on the time of TFA to reach steady state in the river system. The peak surface water TFA
concentrations in the three watersheds all reach steady state relatively quickly within the first year with some small variations,
mainly due to the different flow rates relative to the water volumes in the rivers. The model-predicted surface water TFA
concentrations range from 0.8 ng $L^{-1}$ in High Rhine sub-basin to 3.3 ng $L^{-1}$ in the Delta Rhine sub-basin. The
Rhine sub-basin has higher peak concentrations than High Rhine sub-basin due to lower flow rates and higher suspended
solids. Based on our TFA mass-allocation calculations, approximately 63% of the TFA mass deposited from the air into the
ocean, after the TFA concentrations reached steady state in the surface water and surface soil. According to Sturm et al. (2023),
measurements of TFA in the surface water of the Rhine River are 400 ng $L^{-1}$ near Karlsruhe, Germany (in the Upper Rhine,
sub-basin 3) and 1,100 ng $L^{-1}$ near Mainz, Germany (in the Middle Rhine, sub-basin 4). These measurements suggest that
future pMDIs using NGP as the medical propellant will form negligible TFA (up to 3.3 ng $L^{-1}$, equivalent to less than 1% of
total) in Rhine surface water as compared to other sources that form TFA in the Rhine surface water.

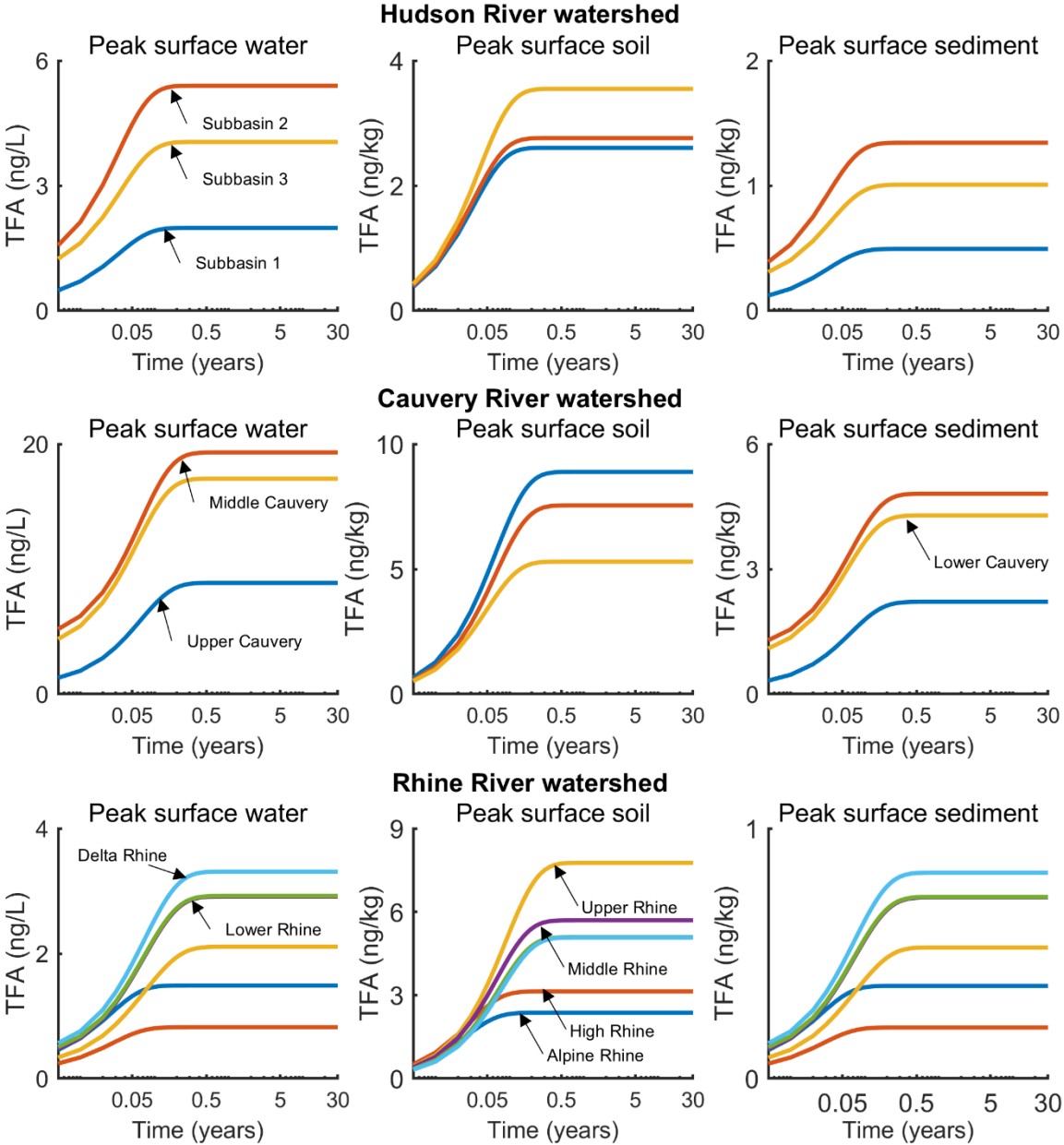

**Figure 7:** Model predicted concentrations of TFA in surface water, surface soil, and surface sediment of Hudson River watershed (first row), Cauvery River watershed (middle row), and Rhine River watershed (bottom row) due to 30-years of continued NGP emissions via pMDI usage.

Because it is not feasible to conduct detailed fate-and-transport analyses for all global watersheds, we instead estimated annual and monthly mean TFA concentrations in rainwater. This was done by dividing the monthly modelled wet deposition flux of TFA by the corresponding monthly modelled precipitation, using GEOS-Chem outputs driven by MERRA-2 reanalysis meteorology. Figure 8 displays the model-predicted annual mean concentrations of TFA in rainwater on a global scale. According to the colour scale, predicted concentrations generally range from about 0.005 to 0.040 µg $L^{-1}$ (5–40 ng $L^{-1}$). Most regions, including North America, Europe, and Asia, show values toward the lower

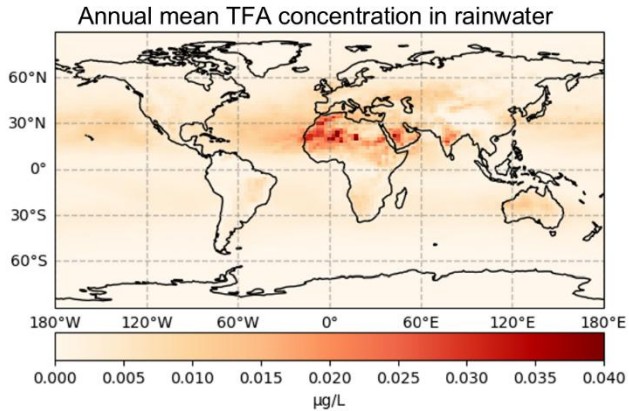

**Figure 8**: Global annual TFA rainwater concentrations computed by dividing the wet TFA deposition flux by MERRA-2 driven predicted precipitation.

end of this spectrum, while some arid areas in North Africa and the Middle East exhibit locally elevated concentrations approaching 40 ng $L^{-1}$. Note that even though arid regions, such as North Africa and the Middle East, have higher TFA rainwater concentrations, the net TFA deposition in these regions will be lower due to low precipitation (Vet et al., 2014). To contextualize these findings, we next compared these values with that reported in the literature and found that even the highest modelled concentrations are several folds lower than the observed values. For example, measured rainwater TFA in North America is frequently within 20–3,779 ng $L^{-1}$ (Garavagno et al., 2024), in Europe (including Germany) medians are between 69 and 350 ng $L^{-1}$ (Freeling et al., 2020), and in Asia (China and Japan) measurements as high as 40–8,800 ng $L^{-1}$ have been reported (Garavagno et al., 2024). In the supplement (Section S3), we have presented the monthly modelled mean TFA rainwater concentrations providing a more granular, month-wise rainwater TFA values that are also in agreement with these annual rainwater TFA comparison.

## 4. Conclusions

Inhaled respiratory medications delivered by pMDIs are relied upon by millions of patients, accounting for 78% of inhaler usage globally (Bell et al., 2023). Due to their high GWP, present-day medical propellants (HFCs) are subject to phase down, under the Kigali amendment. In this study, we estimated environmental TFA accumulation due to prospective usage of pMDIs that have HFO-1234ze(E), as the medical propellant which is not subject to phase-down under the Kigali amendment due to its low GWP. In this study, we investigated atmospheric and surface behavior of HFO-1234ze(E) and its atmospheric breakdown product TFA, focusing on deposition and transport of TFA within the Hudson, Rhine, and Cauvery River watersheds due to continued usage of these prospective pMDIs.

Based on IQVIA MIDAS® annual pMDI volume sales data from all manufacturers with HFO-1234ze(E) as their sole medical propellant, we estimate annual global propellant emissions of 4.736 Gg $yr^{-1}$, concentrated in densely populated regions. As per

Madronich et al. (2023), HFC-134a and HFO-1234yf are the two main sources of TFA in the environment. Their combined annual production rate of TFA is 0.04 – 0.06 Tg yr$^{-1}$. Considering our estimates of annual pMDI-associated propellant emissions and theoretical TFA yield (Tewari et al., 2023), the global total TFA deposition due to future pMDI usage would be ~0.0002 Tg yr$^{-1}$, i.e., $4.736 \times 0.04 \times 10^{-3}$, which suggests that propellant emissions based TFA production represents less than 0.5% of the annual global TFA in the environment estimated by Madronich et al. (2023). Using the GEOS-Chem atmospheric model, adapted for MCM of HFO-1234ze(E), we simulated the effect of prospective pMDI usage, i.e., HFO-1234ze(E) emissions, on environmental TFA deposition patterns. Next, watershed-specific models that follow USEPA guidelines were used to estimate TFA concentrations over 30 years, showcasing how atmospheric TFA may spreads from rivers into the ocean following due to continued pMDI usage. Our study reveals that atmospheric TFA may deposit in soil and water bodies remote to pMDI usage, owing to its mobility and solubility. Atmospheric and watershed modeling predicts that future use of pMDIs may lead to TFA concentrations between 0.8 and 19.3 ng L$^{-1}$ in surface waters, 2.3 and 8.8 ng kg$^{-1}$ in surface soils, and 0.2 and 4.8 ng kg$^{-1}$ in surface sediments across the three studied watersheds. These variations reflect local factors such as water flow, region-specific deposition rates, pMDI usage patterns, and weather conditions.

Lastly, the model's predicted TFA levels can be effectively evaluated by comparison with several established reference values. The highest surface water TFA concentrations attributable to pMDI use are more than 500-fold lower than the German Environment Agency's conservative drinking water threshold of 10,000 ng L$^{-1}$ (Arp et al., 2024). Similarly, these modeled concentrations are greater than 100 times below the Netherlands' most recent drinking water guideline of 2,200 ng L$^{-1}$ (Arp et al., 2024), derived based on precautionary potency factors for PFAS. When placed in ecological context, the maximum TFA level estimated in surface water is also over 6,000 times below the frequently cited no-observed-effect concentration (NOEC) of 120,000 ng L$^{-1}$ for sensitive freshwater algae (Arp et al., 2024), indicating negligible risk to aquatic biota. For soils, predicted TFA loadings from pMDI emissions remain at least 90,000 times lower than the REACH long-term NOEC of 830,000 ng kg$^{-1}$ for plant health (Arp et al., 2024). Furthermore, given that the lowest TFA concentration empirically measured in Rhine surface water is 400 ng L$^{-1}$ (Sturm et al., 2023), prospective new emissions from pMDI use would represent less than 1% of the total TFA present in this major watershed (or catchment).

Taken together, these findings demonstrate that even if HFO-1234ze(E) were to become the sole medical propellant in future pMDIs of all manufacturers, its continual atmospheric release would result in only very low additional quantities of TFA in both surface water and soil—levels that are several orders of magnitude below thresholds for human health or ecological risk. The maximum TFA concentrations projected by this study remain much lower than all currently relevant drinking water, aquatic, and agro-environmental benchmarks. This demonstrates a substantial margin of safety, underscoring that, while TFA is environmentally persistent, its contribution from next-generation propellant use is expected to remain well within safe regulatory and ecotoxicological limits.

## 5. Limitations of the Study

One limitation of the current study is that HFO-1234ze(E) emissions used in the atmospheric modelling of TFA are based on pMDI volume sales data from a single year (2022), which does not account for the projected increase in respiratory disease patients in the future. Emissions of species other than HFO-1234ze(E) were also held constant over the 30-year period. Future trends, particularly in NOx emissions, are expected to influence the yield of TFA due to changes in gas-phase chemistry. The $0.1° \times 0.1°$ spatial resolution of the emissions data may not capture fine-scale variations (<11 km). However, since the smallest sub-basin has an area of approximately 7,040 km², this grid cell size is appropriate for the study's needs. Additionally, the limited geographical coverage of the emissions data may result in an incomplete representation of global HFO-1234ze(E) emissions, potentially affecting the accuracy of TFA formation and transport predictions on a broader scale.

Despite global coverage, the model's standard resolution ($2° \times 2.5°$ with 47 vertical levels) presents some uncertainty, as it may not capture fine-scale variations in chemistry and deposition processes, particularly in regions with complex topography or near emissions sources. TFA deposition estimates made with GEOS-Chem are subject to uncertainties arising from both the meteorological input data and model configuration.

The use of MERRA-2 meteorological inputs introduces uncertainty in projecting future scenarios, mainly due to the complexities of predicting future weather patterns associated with climate change. The chemical mechanism, implemented using KPP version 3.0 and including new chemical reactions for the MCM of HFO-1234ze(E), aims to represent our current understanding of complex atmospheric processes. Despite its detail, there are still uncertainties regarding reaction rate constants, product yields, and possible reaction pathways. Furthermore, to improve computational efficiency, we did not dynamically simulate atmospheric concentrations of $H_2O$, OH, and $HO_2$; instead, the model reads these from the meteorological input (MERRA-2). As a result, omitting them from the reactions does not affect their concentrations in the model, though it may have a minor impact on the spatial pattern of oxidation products.

To estimate the effects of prospective pMDI usage on surface water TFA, we modelled three different watershed regions. In scenarios lacking site-specific data, we used default modelling parameters reflecting average watershed conditions; thus, these may not always accurately represent specific water bodies. However, the impact of these defaults is usually small or insignificant, and most parameter estimates are based on generally available sub-basin data. For example, the empirical intercept coefficient's default value, derived from studies of various watersheds, minimally affects model results but may slightly over- or underestimate TFA concentrations. Since other watershed regions will have different emissions and properties, these results may not be generalizable beyond the specific regions studied. Finally, our conclusions that model-predicted TFA levels do not pose a threat to human health, or the environment are based on currently known safety thresholds. Since information related to TFA toxicity is still evolving, these conclusions may change in the future.

**Code/Data availability**

The original contributions presented in the study are included in the article and the supplement. Additional data and information are available from the corresponding author upon reasonable request.

**Author contribution**

SGT designed the study and prepared the draft. SGT, KV, KZ, LMD, KT, FK, YZ, BY, and CH performed the data analysis.
All authors edited the manuscript.

**Competing interests**

SGT, PG, HK, MG, and SP are employees of AstraZeneca and hold shares/share options in AstraZeneca. KV, KZ, LMD, KT, FK, YZ, and BY are employees of Ramboll. DKP is an employee of Honeywell.

**Acknowledgements**

The authors thank Michael E. Jenkin for developing the Master Chemical Mechanism of HFO-1234ze(E) and preparing the artwork illustrating its atmospheric degradation pathways; Zongrun Li and Viral Shah for their valuable assistance in incorporating photolysis reactions in GEOS-Chem; and Nigel Budgen for assisting with the global monthly pMDI volume
sales data that were used for estimating world-wide HFO-1234ze(E) emissions. The pMDI volume sales data was obtained under license from the following information service: IQVIA MIDAS® monthly volume sales for the period 2022, for 51 countries, for inhalers across all respiratory conditions, reflecting estimates of real-world activity. Copyright IQVIA. All rights reserved, IQVIA market research information is proprietary to IQVIA and available on a confidential basis by subscription from IQVIA.

**Disclaimer**

IQVIA did not provide any support for the analysis or interpretation of the data. The statements, findings, conclusions, views, and opinions expressed are solely those of the authors and are not necessarily those of IQVIA.

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
