# Peer review of "Atmospheric and Watershed Modelling of Trifluoroacetic Acid from Oxidation of HFO-1234ze(E) Released by Prospective Pressurized Metered-Dose Inhaler Use in Three Major River Basins"

_EGUsphere, 2025_

## Referee Comment (RC1)

**Atmospheric and watershed modelling of HFO-1234ze(E) emissions from prospective pressurized metered-dose inhalers usage** by Shivendra G. Tewari et al. MS Preprint egusphere-2025-1031

This paper investigates the production of trifluoroacetic acid (TFA) from the atmospheric oxidation of a latest generation hydrofluoroolefin (HFO), HFO-1234ze(E), that is being used as a replacement for previous generation propellants in pressurised metered-dose inhalers (pMDI). The paper also investigates the subsequent TFA deposition within three separate river drainage basins, produced from the atmospheric oxidation of HFO-1234ze(E). The paper uses well-tested methods and models but I have several concerns (outlined below) with regards to the oxidation pathways, data used and conclusions made. The paper is interesting and scientifically valuable as not many studies have been done on pMDI emissions; therefore, once the following issues have been addressed, the paper merits publication in ACP.

**General comments**

Based on the title of the paper, I would expect to see more information on other atmospheric oxidation routes of HFO-1234ze(E). Despite OH being the most probable oxidation pathway for this HFO due to the reaction rate, ozonolysis has been shown as a viable oxidation pathway to ultimately produce trifluoromethane (HFC-23; $CHF_3$) with yields of 3.11 ± 0.05% (https://doi.org/10.1073/pnas.2312714120) despite the lower reaction rates. As pMDI emissions are likely to occur in the lower troposphere, un regionally polluted areas with high ozone, ozonolysis will play a role in the oxidation of HFC-1234ze(E). The authors allude to another oxidation process, i.e. photolysis, of trifluoroacetyl aldehyde (TFAA) that does not produce TFA; however, it does produce HFC-23. Considering the main focus of this paper is the atmospheric breakdown of HFO-1234ze(E), these mechanisms that produce other trace gases should be outlined at least in the introduction, irrespective of if the focus is only on TFA production.

The study uses nitric oxide (NO) emission estimates as a proxy for the spatial distribution of HFO-1234ze(E), which are then scaled based on the monthly pMDI sales from the top 51 countries to calculate total emissions per month per country. A significant limitation of this methodology does not account for biogenic NO emissions (i.e. partial denitrification), aircraft or ship emissions, which would not be associated with pMDI usage. Rather than using NO as a proxy, a priori information of population density or nighttime light could be used to better resolve emissions distributions. Furthermore, the full inclusion of information used to estimate emissions of HFO-1234ze(E) from the use of pMDI should be included. Currently no sales data are included in the paper or supplement, making it impossible for the reader to recreate the results given in the paper. Additionally, the limitation of only using sales data from one pharmaceutical company reduces the representativeness of the results. It would be better to use total global sales of pMDIs instead. If the latter has been used, it was not made clear within the paper.

The conclusion that even if HFO-1234ze(E) was the only propellant used in pMDIs, the continuous emission to the atmosphere would have negligible effects on surface water and soil TFA concentrations is misleading. Albeit that the results from this study might

indicate that this is the case, they are based on a single emission rate of HFO-1234ze(E) that does not vary over time and on sales data from one single pharmaceutical company. Considering the total number of asthma and COPD diagnoses continue to increase globally (https://doi.org/10.1586/17476348.2016.1114417), a time varying emission estimate would be more representative, as well as using global sales data.

**Specific comments**

Lines 1-2: As previously outlined in the general comments, the title is not representative of the true focus of the paper and is too broad. Greater specificity is needed so that a reader truly knows what the paper is about. A suggested title, based on the current scope, is "Atmospheric and watershed modelling of trifluoroacetic acid from the oxidation of HFO-1234ze(E) from pressurized metered-dose inhaler usage".

Lines 13 – 15: Halogenated hydrocabons with a '—CF$_3$' moiety also have the possibility to produce HFC-23. Despite this not being within the scope of the paper, this pathway should at least be acknowledged.

Despite the ACP house style specifically stating that units must be written exponentially (i.e. Gg yr$^{-1}$), the authors have not done this throughout the paper.

Lines 20 - 22: The authors assume that the model-predicted TFA deposition rates, which are higher in tropical regions than northern-temperate regions despite higher sales of pMDI in the latter region, is based on photolysis of TFAA being the dominant process; however, this assumption does not seem to take into account atmospheric transport and the lifetime of HFO-1234ze(E). Moreover, this could be a function of the spatial distribution estimation using NO emissions (see General comments). Have the authors looked into a sensitivity analysis to verify if the photolysis reaction of TFAA is dominant in northern-temperate zones?

Watershed is a contentious word within hydrology as it can have different meanings depending on readers nationality (c.f. river drainage basin as a unit, line that separates neighbouring drainage basins, a time of day in broadcasting). To avoid confusion and increase readership beyond that of North America, it would be good to either change the word to a less contentious naming or have a definition of what is meant by watershed.

Lines 26 – 28: It might be that pMDI usage-based HFO-1234ze(E) emissions only do not pose a risk to human health based on the results and data used in this study but they contribute to other sources. It would be good to acknowledge that the culmination in other sources and growth in pMDI usage might change this, as well as contextualising with the limitations of the paper.

Lines 38 – 40: As the scope of the paper is global, it would be good to include other non-USA regulations on the phase out of HFCs too, i.e. European F-gas regulation.

Line 80 – 81: Why were emissions calculate at 30-minute time steps as this must have made the running of the model computationally expensive? As the anthropogenic and biogenic emission estimates outlined in section 2.2 that are used in the model have a temporal resolution of one month, why was the 30-minute time step chosen?

Lines 95 – 112: How come the method for HFO-1234ze(E) emissions calculations are not outlined in this section? It would be a pertinent location to include it to help any reader find the necessary information to run GEOS-Chem to verify your results. It would also allow the authors to give more information on the sales data used (c.f. one company vs. global sales values) and how this was scaled. Currently very little information is given on how these emissions are calculated and this is crucial information to be able to replicate the results.

Lines 132 - 134: The exclusion of $O_2$ and $H_2O$ from GEOS-Chem reactions is significant limitation in the study as atmospheric mole fractions of these vary spatially and temporally and will have an impact on the different reactions associated with the oxidation of HFO-1234ze(E). For example, $O_2$ is required in the $CF_3CH(OH)CHF(O^\bullet)$ pathway to produce TFAA (see figure 5 in Behringer et al., 2021).

Lines 226 – 229: What was the rationale for choosing the southern Indian drainage basin? India is an Article 5 party under the Montreal Protocol and its Amendments; therefore, it has a delayed phase-down of HFCs compared to non-article 5 countries (i.e. western countries such as USA or Europe). As such, the use of HFCs in pMDI would be expected to continue on longer in India than the USA or Europe.

Lines 232 – 235: How do you account for the granularity in GEOS-Chem spatial resolution and the resultant TFA deposition rates (model resolution of 2° x 2.5°) on the deposition rates put into the drainage basin model?

Lines 256 – 268: How does the model account for infiltration and percolation rates across the river drainage basins, as these will vary spatially based on soil and bedrock types.

Lines 271 – 281: The results and discussion section should not be where the method of distributing HFO-1234ze(E) emissions from pMDI usage I located. Instead, it should be included in section 2.2. Furthermore, the monthly pMDI sales data used within the paper should be included, either in figure or tabulated form, so that readers of the paper can use the data along with other methods outlined in the paper to verify the results presented in the paper.

Lines 287 – 291: Considering that NO is used as a proxy and often total emissions of this gas do not vary significantly temporally, would one not expect to see the same in the HFO-1234ze(E) emissions considering NO is used as an emissions proxy? Furthermore, based on your methodology, I don't believe you can current definitively state that there are sustained demand of pMDIs in the respiratory disease community without temporal variations.

Lines 301 – 302: How does TFA deposition flux vary seasonally? It would be good to describe it here in the main text, despite not requiring the figures in the main text.

Lines 302 – 206: See Specific Comment for Lines 20 – 22, regarding transport of TFA and the subsequent dry deposition.

Lines 334 – 336: The description of the model and compartmentalisation should be included in the methods section.

Figure 6: The GEOS-Chem gridded TFA deposition over the selected river drainage basins often bridge both the drainage basin and the surrounding area. How are these deposition rates accounted for in the sections where the gridded deposition has both regions contained within it?

Lines 340 – 343: Why are the authors assuming the TFA deposition fluxes do not change temporally? Does Fig. S3 and S4 not show that these do vary both temporally and spatially for dry and wet deposition, respectively.

Lines 404 – 414: I find these three points hard to read and believe that they should be rewritten. Additionally, I find the language used to be quite emotive and political. Rather than pushing an agenda, results should be presented scientifically and then contextualised, but taking the assumptions and limitations of the study into account.

Lines 416 – 420: As for the previous point, this whole paragraph is pushing an agenda, whereas it would be better to outline that these results are from sales figures from a single pharmaceutical company and that emissions of HFO-1234ze(E) are not limited to this one source or company. Instead, it contributes to total emissions.

**Technical corrections**
Line 13: HFO is not defined prior to its first use.

Line 16: What is environmental TFA? Suggest removing "environmental" as it is unnecessary.

Line 36: Please use a more up-to-date $GWP_{100}$ value for HFC-134a. The Annex to the WMO Scientific Assessment of Ozone Depletion 2022 has a more recent value of 1,470.

Line 41: HFO-1234ze(E) wasn't purely developed by the pharmaceutical industry as a propellant and is currently being used as a refrigerant and foam blowing agent. Emissions are already observed globally by the AGAGE network (see https://doi.org/10.1021/es505123x). Please reword this sentence.

Lines 49 – 52: As far as I am aware, the PFAS regulation was passed in the EU and should be coming into law soon. Please update this sentence to reflect this.

Lines 62 – 65: Please outline the other TFAA degradation pathays that have been suggested in the literature. From the literature cited, it looks like the authors are mostly

suggesting the photolysis of TFAA to produce HFC-23 but it is not entirely clear. There are pathways of TFAA oxidation through hydrolysis that can produce TFA – see Figure 8 in Berhringer et al (2021; https://www.umweltbundesamt.de/sites/default/files/medien/5750/publikationen/2021-05-06_texte_73-2021_persistent_degradation_products.pdf) for more information.

Line 65 – 66: The assertion that no prior work has explored TFA production from HFO-1234ze(E) is slightly misleading. The authors have produced a similar study using a box-model with chemical degradation mechanisms included (see https://doi.org/10.3389/fenvs.2023.1297920).

Lines 75 – 81: Please include that GEOS-Chem is a Eularian model.

Line 119: If stating the authors name in the text, it does not then need to be included in the citation in parenthesis – "... in GEOS-Chem by Mao et al. (2010) for the ...". See the ACP house style for more information (https://www.atmospheric-chemistry-and-physics.net/submission.html#references)

Lines 139 - 140: This sentence is not complete.
Line 375: Replace "Alphine" with "Alpine".

Line 389: state what the $GWP_{100}$ For HFO-1234ze(E) is.

Fig. S6: Correlation values are usually stated as an R value in statistics, whereas p-values are the statistical significance of each test. Both should be reported.

---

## Author Comment (AC1)

**Atmospheric and watershed modelling of HFO-1234ze(E) emissions from prospective pressurized metered-dose inhalers usage** by Shivendra G. Tewari et al.

MS Preprint egusphere-2025-1031

This paper investigates the production of trifluoroacetic acid (TFA) from the atmospheric oxidation of a latest generation hydrofluoroolefin (HFO), HFO-1234ze(E), that is being used as a replacement for previous generation propellants in pressurized metered-dose inhalers (pMDI). The paper also investigates the subsequent TFA deposition within three separate river drainage basins, produced from the atmospheric oxidation of HFO-1234ze(E). The paper uses well-tested methods and models but I have several concerns (outlined below) with regards to the oxidation pathways, data used and conclusions made. The paper is interesting and scientifically valuable as not many studies have been done on pMDI emissions; therefore, once the following issues have been addressed, the paper merits publication in ACP.

**Response**: We thank the Reviewer for their positive feedback. In the revised manuscript, we have addressed all of their concerns. Below, we provide our point-by-point responses to each issue raised.

**General comments**

Based on the title of the paper, I would expect to see more information on other atmospheric oxidation routes of HFO-1234ze(E). Despite OH being the most probable oxidation pathway for this HFO due to the reaction rate, ozonolysis has been shown as a viable oxidation pathway to ultimately produce trifluoromethane (HFC-23; CHF3) with yields of 3.11 ± 0.05% (https://doi.org/10.1073/pnas.2312714120) despite the lower reaction rates. As pMDI emissions are likely to occur in the lower troposphere, un regionally polluted areas with high ozone, ozonolysis will play a role in the oxidation of HFC-1234ze(E). The authors allude to another oxidation process, i.e. photolysis, of trifluoroacetyl aldehyde (TFAA) that does not produce TFA; however, it does produce HFC-23. Considering the main focus of this paper is the atmospheric breakdown of HFO-1234ze(E), these mechanisms that produce other trace gases should be outlined at least in the introduction, irrespective of if the focus is only on TFA production.

**Response**: The Reviewer raises two points related to (1) the removal of HFO-1234ze(E) by reaction with ozone and (2) the photolysis of trifluoroacetaldehyde (TFAA), and the reported formation of HFC-23 from each of these routes.

1) Reaction of HFO-1234ze(E) with ozone

All tropospheric removal routes for HFO-1234ze(E) were considered in construction of the chemical mechanism. As the Reviewer indicates, removal by reaction with OH is the dominant removal route, the 298 K rate coefficient ($k_{OH}$) currently recommended by the IUPAC Task Group on Atmospheric Chemical Kinetic Data Evaluation being $7.0 \times 10^{-13}$ cm$^3$ molecule$^{-1}$ s$^{-1}$. A widely-applied criterion for inclusion of the reaction of an organic compound with ozone in chemical mechanisms such as the MCM (e.g. Jenkin et al., 1997) is if its rate coefficient ($k_{O3}$) is greater than $10^{-8} k_{OH}$, this being the threshold where about 1% of the organic compound is estimated to be removed by reaction with ozone. The currently recommended rate coefficient is $k_{O3} = 2.5 \times 10^{-21}$ cm$^3$ molecule$^{-1}$ s$^{-1}$ (IUPAC Task Group on Atmospheric Chemical Kinetic Data Evaluation – Data Sheet oFOx129), such that $k_{O3} = 3.6 \times 10^{-9} k_{OH}$. Reaction with ozone was therefore not represented.

The Reviewer comments on the relatively recent study of McGillen et al. (2023) which reports a small yield (about 3%) of HFC-23 from the slow reaction of HFO-1234ze(E) with ozone, and a resultant increase in the GWP calculated for HFO-1234ze(E) (i.e. from about 1 to about 12 on the 100-year

horizon). Although this is an important result, it is possible that the impact has been overestimated. First, McGillen et al. (2023) report that about 3% of HFO-1234ze(E) is removed by reaction with ozone in their global modelling study, the highest fraction of all the HFOs considered. Based on the relative OH and ozone rate coefficients reported in their study, it seems likely that the highest fractional removal by ozone should be for HFO-1243zf, with that for HFO-1234ze(E) being somewhat lower (as discussed above). Secondly, McGillen et al. (2023) applied temperature independent rate coefficients for the HFO + ozone reactions in their model. In practice, these reactions are expected to possess strong temperature dependences. Based on the reaction of ozone with cis-dichloroethene (which is of comparable reactivity to HFO-1234ze(E)), the rate coefficient would be expected to fall by a factor of about 100 between 298K and an upper troposphere temperature of 220K (e.g. Leather et al. 2011, this study being from the same group as McGillen et al. (2023)). Representation of the likely temperature dependence would therefore further reduce the fractional loss of HFO-1234ze(E) by reaction with ozone, and the calculated HFC-23 yield from its global oxidation.

Regarding the possible increased removal by reaction with ozone in locally polluted regions where HFO-1234ze(E) is emitted, it should be noted that even at 100 ppb of ozone at ground level, the lifetime of HFO-1234ze(E) with respect to reaction with ozone is 5 years. Consequently, its removal is not influenced by local ozone hotspots, but by transport and the global distribution of oxidants (specifically OH radicals), with its oxidation mainly occurring in regions where the average OH radical concentrations are the highest.

2) Photolysis of TFAA

The photolysis of TFAA is currently recommended to have only one significant channel under lower atmospheric conditions, namely by C-C bond scission to produce $CF_3$ and HCO  (IUPAC Task Group on Atmospheric Chemical Kinetic Data Evaluation – Data Sheet IV PF3). The further oxidation of these products does not form TFA or HFC-23.

The Reviewer is correct that there is also a product channel forming $CHF_3$ (HFC-23) and CO. This has long been known to contribute at wavelengths shorter than the tropospheric range. The preliminary paper (not peer-reviewed) from 2021 highlighted by the Reviewer (https://doi.org/10.21203/rs.3.rs-199769/v1) reports a contribution from this channel at 308 nm (i.e. within the tropospheric range), but based on observation under low pressure collision-free conditions. A subsequent full paper from the same group (Thomson et al., 2024), confirms a quantum yield of 0.16 at 308 nm for formation of HFC-23 under collision-free conditions, but with this falling to 0.00023 at atmospheric pressure. In the interim Sulbaek-Andersen et al. (2022) and Sulbaek-Andersen et al. (2023) have found no evidence for HFC-23 formation under tropospheric conditions (in agreement with earlier studies, e.g. Chiappero et al., 2006). As a result the IUPAC Task Group recommends a quantum yield of < 0.003 at 308 nm (IUPAC Task Group on Atmospheric Chemical Kinetic Data Evaluation – Data Sheet IV PF3).

Because of the points discussed in 1) and 2), we did not consider HFC-23 formation in the MCM mechanism of HFO-1234ze(E). Although since 1) is an important result, in the revised manuscript, we have added a few sentences stating that formation of HFC-23 from atmospheric breakdown of HFO-1234ze(E) is possible.

The study uses nitric oxide (NO) emission estimates as a proxy for the spatial distribution of HFO-1234ze(E), which are then scaled based on the monthly pMDI sales from the top 51 countries to calculate total emissions per month per country. A significant limitation of this methodology does not account for biogenic NO emissions (i.e. partial denitrification), aircraft or ship emissions, which would not be associated with pMDI usage. Rather than using NO as a proxy, a priori information of population density or nighttime light could be used to better resolve emissions distributions. Furthermore, the full inclusion of information used to estimate emissions of HFO-1234ze(E) from the use of pMDI should be included. Currently no sales data are included in the paper or supplement, making it impossible for the reader to recreate the results given in the paper. Additionally, the limitation of only using sales data from one pharmaceutical company reduces the representativeness

of the results. It would be better to use total global sales of pMDIs instead. If the latter has been used, it was not made clear within the paper.

**Response**: As specified in the manuscript, the NO emissions used as a spatial proxy for HFO-1234ze(E) emissions are from the residential and commercial sectors of the CEDS inventory. This was done intentionally, as these sectors are representative of population density and human activity and are therefore more likely to correlate with pMDI usage. Emissions from unrelated sources—such as biogenic, aircraft, or maritime—were not included in this proxy.

With the revised manuscript, we will include a new supplementary data which contains the country wise HFO-1234ze(E) emissions based pMDI volume sales data. Note that the pMDI data is for all manufacturers across all respiratory diseases. We have revised the relevant sections throughout to clarify this ambiguity.

The conclusion that even if HFO-1234ze(E) was the only propellant used in pMDIs, the continuous emission to the atmosphere would have negligible effects on surface water and soil TFA concentrations is misleading. Albeit that the results from this study might indicate that this is the case, they are based on a single emission rate of HFO-1234ze(E) that does not vary over time and on sales data from one single pharmaceutical company. Considering the total number of asthma and COPD diagnoses continue to increase globally (https://doi.org/10.1586/17476348.2016.1114417), a time varying emission estimate would be more representative, as well as using global sales data.

**Response**: We agree that there are limitations to this study, such as assuming pMDI sales do not increase over time. Note that the monthly pMDI sales data that we have used to calculate global HFO-1234ze(E) is from all manufacturers, i.e., not tied to a single pharmaceutical company. The Reviewer is further correct that numbers of COPD/Asthma patients are projected to increase by 23% in the next thirty years. However, without hard evidence, it is difficult understand what regions will see the most increase of such patients. Moreover, we cannot predict weather patterns for the next thirty years which should also affect TFA formation and deposition globally. Therefore, for relative simplicity, we used region-specific annual TFA deposition flux for calculating net accumulation of TFA in surface water, soil, and sediment over time.

**Specific comments**

Lines 1-2: As previously outlined in the general comments, the title is not representative of the true focus of the paper and is too broad. Greater specificity is needed so that a reader truly knows what the paper is about. A suggested title, based on the current scope, is "Atmospheric and watershed modelling of trifluoroacetic acid from the oxidation of HFO-1234ze(E) from pressurized metered-dose inhaler usage".

**Response**: Thank you for the feedback. We have revised the title of the manuscript to "Atmospheric and Watershed Modelling of Trifluoroacetic Acid from Oxidation of HFO-1234ze(E) Released by Prospective Pressurized Metered-Dose Inhaler Use."

Lines 13 – 15: Halogenated hydrocabons with a '—CF3' moiety also have the possibility to produce HFC-23. Despite this not being within the scope of the paper, this pathway should at least be acknowledged. Despite the ACP house style specifically stating that units must be written exponentially (i.e. Gg yr-1), the authors have not done this throughout the paper.

**Response**: As discussed under the "General comments," we have added a few sentences in the Introduction of the revised manuscript to address this comment. In the revised manuscript, we have also edited the units to match the exponential format of ACP journal.

Lines 20 - 22: The authors assume that the model-predicted TFA deposition rates, which are higher in tropical regions than northern-temperate regions despite higher sales of pMDI in the latter region, is based on photolysis of TFAA being the dominant process; however, this assumption does not seem to take into account atmospheric transport and the lifetime of HFO-1234ze(E). Moreover, this could be a function of the spatial distribution estimation using NO emissions (see General comments). Have the

authors looked into a sensitivity analysis to verify if the photolysis reaction of TFAA is dominant in northern-temperate zones?

**Response**: We would like to clarify that the model-predicted TFA deposition is calculated using the GEOS-Chem chemical transport model, which explicitly accounts for atmospheric processes, including chemical transformation, transport, and both dry and wet deposition. Our conclusion that photolysis of TFAA is dominant is based on the short atmospheric lifetime of HFO-1234ze(E), approximately 20 days. This suggests that the transport of the propellant will have a smaller effect on distant regions compared to other intermediates, such as TFA itself. Since we did not perform a sensitivity analysis in deriving these conclusions, we have updated the relevant section to acknowledge that the impact of TFA transport cannot be completely ruled out.

Additionally, please note that while the spatial distribution of emissions is based on NO from residential and commercial sources, the resulting TFA deposition patterns are determined by the model's comprehensive treatment of atmospheric dynamics and chemistry. This means that although TFA deposition is influenced by NO emissions, the deposition patterns will not directly correlate with the NO emission patterns.

Watershed is a contentious word within hydrology as it can have different meanings depending on readers nationality (c.f. river drainage basin as a unit, line that separates neighbouring drainage basins, a time of day in broadcasting). To avoid confusion and increase readership beyond that of North America, it would be good to either change the word to a less contentious naming or have a definition of what is meant by watershed.

**Response**: In the revised manuscript, we have clarified that "watershed" means drainage basin or catchment area to establish a common understanding of how it is used throughout the paper. As used here, a watershed is an area that channels precipitation and runoff into a common body of water (also sometimes referred to as a drainage basin or catchment).

Lines 26 – 28: It might be that pMDI usage-based HFO-1234ze(E) emissions only do not pose a risk to human health based on the results and data used in this study but they contribute to other sources. It would be good to acknowledge that the culmination in other sources and growth in pMDI usage might change this, as well as contextualizing with the limitations of the paper.

**Response**: We agree with the Reviewer that TFA's effect on the environment are only beginning to be understood. To address these concerns, we have added a new section "Limitations of the Study" in the revised manuscript (see below).

"**Limitations of the Study**
One limitation of the current study is that HFO-1234ze(E) emissions used in the atmospheric modelling of TFA are based on pMDI volume sales data from a single year (2022), which does not account for the projected increase in respiratory disease patients in the future. Emissions of species other than HFO-1234ze(E) were also held constant over the 30-year period. Future trends, particularly in NOx emissions, are expected to influence the yield of TFA due to changes in gas-phase chemistry. The $0.1° \times 0.1°$ spatial resolution of the emissions data may not capture fine-scale variations (<11 km). However, since the smallest sub-basin has an area of approximately 7,040 km², this grid cell size is appropriate for the study's needs. Additionally, the limited geographical coverage of the emissions data may result in an incomplete representation of global HFO-1234ze(E) emissions, potentially affecting the accuracy of TFA formation and transport predictions on a broader scale. TFA deposition estimates made with GEOS-Chem are subject to uncertainties arising from both the meteorological input data and model configuration. The use of MERRA-2 meteorological inputs introduces uncertainty in projecting future scenarios, mainly due to the complexities of predicting future weather patterns associated with climate change. The chemical mechanism, implemented using KPP version 3.0 and including new chemical reactions for the MCM of HFO-1234ze(E), aims to represent our current understanding of complex atmospheric processes. Despite its detail, there are

still uncertainties regarding reaction rate constants, product yields, and possible reaction pathways. Furthermore, to improve computational efficiency, we did not dynamically simulate atmospheric concentrations of $H_2O$, OH, and $HO_2$; instead, the model reads these from the meteorological input (MERRA-2). As a result, omitting them from the reactions does not affect their concentrations in the model, though it may have a minor impact on the spatial pattern of oxidation products.

To estimate the effects of prospective pMDI usage on surface water TFA, we modelled three different watershed regions. In scenarios lacking site-specific data, we used default modelling parameters reflecting average watershed conditions; thus, these may not always accurately represent specific water bodies. However, the impact of these defaults is usually small or insignificant, and most parameter estimates are based on generally available sub-basin data. For example, the empirical intercept coefficient's default value, derived from studies of various watersheds, minimally affects model results but may slightly over- or underestimate TFA concentrations. Finally, our conclusions that model-predicted TFA levels do not pose a threat to human health or the environment are based on currently known safety thresholds. Since information related to TFA toxicity is still evolving, these conclusions may change in the future."

Lines 38 – 40: As the scope of the paper is global, it would be good to include other non-USA regulations on the phase out of HFCs too, i.e. European F-gas regulation.

**Response**: In the revised manuscript, we have added that HFCs are being phased out in Europe under European F-gas regulation as well.

Line 80 – 81: Why were emissions calculate at 30-minute time steps as this must have made the running of the model computationally expensive? As the anthropogenic and biogenic emission estimates outlined in section 2.2 that are used in the model have a temporal resolution of one month, why was the 30-minute time step chosen?

**Response**: The 30-minute emissions time step refers to how frequently the emissions fields are read and updated within the GEOS-Chem model, and it aligns with the model's default configuration. While the anthropogenic and biogenic emissions used in this study are provided at monthly resolution, GEOS-Chem internally applies temporal scaling factors (e.g., diurnal, weekly) where available to generate more realistic sub-monthly variability. Since the model was run at a spatial resolution of 2° × 2.5° on a high-performance Linux system configured to handle parallel processing similarly to an HPC cluster, the computational cost associated with using a 30-minute emissions update frequency was minimal and did not present a significant burden.

Lines 95 – 112: How come the method for HFO-1234ze(E) emissions calculations are not outlined in this section? It would be a pertinent location to include it to help any reader find the necessary information to run GEOS-Chem to verify your results. It would also allow the authors to give more information on the sales data used (c.f. one company vs. global sales values) and how this was scaled. Currently very little information is given on how these emissions are calculated and this is crucial information to be able to replicate the results.

**Response**: We have now included the following description of emissions in Section 2 of the revised manuscript. Specifically, monthly pMDI sales data from all manufacturers for the top 51 countries were used to estimate emissions, assuming a daily dose of 4 puffs/day and ~14 g of HFO-1234ze(E) per pMDI. These emission estimates were converted to monthly totals and spatially distributed within each country using NO emissions from the residential and commercial sectors in the CEDS inventory as a proxy. This approach assumes that NO emissions in these sectors correlate with population density and, hence, pMDI usage. Consequently, the spatial resolution of the emissions follows that of the CEDS proxy data. With the revised manuscript, we will now provide a supplement data containing our estimates of global monthly HFO-1234ze emissions for the top 51 countries.

Lines 132 - 134: The exclusion of O2 and H2O from GEOS-Chem reactions is significant limitation in the study as atmospheric mole fractions of these vary spatially and temporally and will have an impact

on the different reactions associated with the oxidation of HFO-1234ze(E). For example, $O_2$ is required in the $CF_3CH(OH)CHF(O\bullet)$ pathway to produce TFAA (see figure 5 in Behringer et al., 2021).

**Response**: $O_2$ is indeed involved in many tropospheric reactions, with a large number of organic radicals reacting rapidly and often exclusively with $O_2$. Because these reactions occur essentially instantaneously (owing to the high concentration of $O_2$), and there is no competing reaction in many cases, it is not necessary to represent the reaction explicitly in these cases, and the reaction is invariably combined into the preceding reaction in tropospheric chemical mechanisms for computational efficiency.

This is the case for many reactions in the HFO-1234ze(E) degradation mechanism. For example, the addition reaction of OH with HFO-1234ze(E) initially forms two possible organic radicals, each combining rapidly with $O_2$ to form a peroxy radical, e.g.,

$OH + CF_3CH=CHF \rightarrow CF_3CH(OH)\dot{C}HF$ (1)

$CF_3CH(OH)\dot{C}HF + O_2\ (+M) \rightarrow CF_3CH(OH)CH(O\dot{O})F\ (+M)$ (2)

Because of the high concentration of $O_2$, reactions of type (2) typically occur on the timescale of ≤ 25 ns in air at atmospheric pressure (i.e., instantaneously), such that reaction (1) is the rate-determining step – the timescale being several weeks. The reactions can therefore be combined, and it is not necessary to represent the involvement of $O_2$ (or M) explicitly, i.e.:

$OH + CF_3CH=CHF \rightarrow CF_3CH(OH)CH(O\dot{O})F$ (1) + (2)

In the example given by the reviewer, as illustrated in the report of Behringer et al. (2021), the subsequent reaction of this peroxy radical with NO forms an oxy radical, $CF_3CH(OH)CH(\dot{O})F$, which initially decomposes as follows:

$CF_3CH(OH)CH(O\dot{O})F + NO \rightarrow CF_3CH(OH)CH(\dot{O})F + NO_2$ (3)

$CF_3CH(OH)CH(\dot{O})F\ (+M) \rightarrow CF_3\dot{C}HOH + HC(O)F\ (+M)$ (4)

The radical product, $CF_3\dot{C}HOH$, then reacts rapidly and exclusively with $O_2$:

$CF_3\dot{C}HOH + O_2 \rightarrow CF_3CHO + HO_2$ (5)

(Note: the report of Behringer et al. (2021), and the source paper of Javadi et al. (2008), incorrectly show OH as the co-product of $CF_3CHO$).

In this reaction sequence, reaction (3) is the rate determining step, and neither reaction (4) nor reaction (5) needs to be represented because they occur very rapidly and have no competing processes. The combined reaction does not then contain $O_2$ explicitly, although its involvement is included:

$CF_3CH(OH)CH(O\dot{O})F + NO \rightarrow HC(O)F + CF_3CHO + HO_2 + NO_2$   (3)+(4)+(5) *(note this is Reaction R2 in the manuscript)*

Methods like those described above are commonly used to make tropospheric chemical mechanisms more computationally efficient, as the involvement of $O_2$ is intrinsically incorporated into many reactions represented in the applied mechanism. Moreover, both $O_2$ and $H_2O$ are included in the core GEOS-Chem model, provided by the meteorological input data (MERRA2), and are not dynamically simulated within the chemical mechanism itself. Consequently, although $O_2$ and $H_2O$ were not explicitly added to the TFA chemical mechanism (as in the $O_2$ example above), their concentrations in the model are unaffected. Thus, omitting them from the TFA mechanism reactions does not impact their modelled concentrations. However, this omission could have a minor influence on the spatial distribution of oxidation products as noted in the new "Limitations of the study" section. For completeness, in the revised manuscript and supplement, we have updated the HFO-1234ze(E) MCM reactions to include $O_2$ and $H_2O$ wherever these species are necessary to balance the reactions.

Lines 226 – 229: What was the rationale for choosing the southern Indian drainage basin? India is an Article 5 party under the Montreal Protocol and its Amendments; therefore, it has a delayed phase-down of HFCs compared to non-article 5 countries (i.e. western countries such as USA or Europe). As such, the use of HFCs in pMDI would be expected to continue on longer in India than the USA or Europe.

**Response**: The Reviewer correctly notes that the phasedown of HFCs in India will take longer than in the US or Europe. In this study, we planned to investigate TFA levels in the surface waters of three watersheds. We chose the Rhine watershed in Europe as the primary location due to the availability of numerous TFA-related literature references for comparison. The US was selected because it has the highest pMDI sales/usage globally. India was chosen as the third watershed because it ranks as the third-largest pMDI market in the world, after the US and the UK. Since we were already considering the Rhine watershed (Europe), we decided to include India instead of the UK as the third location. In the revised manuscript, we have briefly added this rationale for selecting three watersheds.

Lines 232 – 235: How do you account for the granularity in GEOS-Chem spatial resolution and the resultant TFA deposition rates (model resolution of 2° x 2.5°) on the deposition rates put into the drainage basin model?

**Response**: The modeled deposition dataset was superimposed onto basin boundaries in GIS to estimate TFA deposition flux in each subbasin of each basin. The TFA deposition flux for each subbasin was calculated using the gridded dataset by computing a weighted average that accounts for the overlapping area of each grid cell with the subbasin. The 2° x 2.5° resolution of the TFA deposition data does not fully capture granularity, specifically in smaller subbasins, but this resolution is necessary considering the global scale of the modeling and is typical of such modeling.

Lines 256 – 268: How does the model account for infiltration and percolation rates across the river drainage basins, as these will vary spatially based on soil and bedrock types.

**Response**: The infiltration and percolation rates are primarily affected by precipitation, irrigation, run-off, and evapotranspiration in this model. This study is a conceptual level study and does not account for micro-scale variations such as soil types and bedrock types which could be specific for a local area. Modeling parameters such has precipitation was estimated for each sub-basin and accounted for spatial variation. When sub-basin specific data were not readily available for parameters such as irrigation, run-off, and evapotranspiration, they were based on as average values across the entire Rhine River basin. Using average values for the entire basin could have some impact on the modeling results for each sub-basin but should not have significant impact on the results (i.e., changing the order of magnitude of the results).

Lines 271 – 281: The results and discussion section should not be where the method of distributing HFO-1234ze(E) emissions from pMDI usage I located. Instead, it should be included in section 2.2. Furthermore, the monthly pMDI sales data used within the paper should be included, either in figure or tabulated form, so that readers of the paper can use the data along with other methods outlined in the paper to verify the results presented in the paper.

**Response**: The Reviewer raises a valid point regarding the placement of emission information in the Methods section. However, we believe that our global pMDI emission calculations are novel and also merits inclusion in the Results section. To ensure proper flow of information, we have added a few sentences to outline how the emissions were calculated and have provided an indication for readers that further details are available in the Results section. We will include the global, monthly propellant emission data as a supplement of the revised manuscript.

Lines 287 – 291: Considering that NO is used as a proxy and often total emissions of this gas do not vary significantly temporally, would one not expect to see the same in the HFO-1234ze(E) emissions considering NO is used as an emissions proxy? Furthermore, based on your methodology, I don't believe you can current definitively state that there are sustained demand of pMDIs in the respiratory disease community without temporal variations.

**Response**: We thank the Reviewer for raising this important point. Our conclusion that pMDI-based HFO emissions do not vary seasonally is based on the global monthly average of propellant emissions (Fig. S2), which shows only minimal changes in pMDI-based HFO emissions over time.

Following the Reviewer's suggestion, we examined region-specific global, monthly propellant emission (pMDI sales) data. We observed that propellant emissions do indeed vary in some regions—for example, in the UK and Brazil, as shown in the new supplemental data show estimated propellant emissions (also, see below). However, the timing of peak emissions (pMDI sales) differs between regions. When these region-specific differences are averaged globally (as in Fig. S2), the variations are largely offset and thus not apparent at the global level.

We have clarified this reason, in the revised manuscript, explaining why minimal variation in pMDI-based HFO emissions is observed over time.

[Figure]

Lines 301 – 302: How does TFA deposition flux vary seasonally? It would be good to describe it here in the main text, despite not requiring the figures in the main text.

**Response**: TFA tends to have higher deposition rates in the summer as refrigerants are more commonly used in warmer months and more oxidants are present in summer, although precipitation is also a factor as it allows for more wet deposition. Wang et al. (2018) noted that maximum wet and dry TFA deposition occurred during the summer, accounting for 71% of global yearly deposition, followed by autumn with 12% global yearly deposition. Wang et al. (2018) also stated that although HFO-1234yf emissions are the dominant factor for TFA deposition, convective mass flux and precipitation also have an effect.

Lines 302 – 206: See Specific Comment for Lines 20 – 22, regarding transport of TFA and the subsequent dry deposition.

**Response**: In the revised manuscript, we have added that higher deposition of TFA around the tropical region can also be due to transport of TFA from other regions.

Lines 334 – 336: The description of the model and compartmentalisation should be included in the methods section.

**Response**: We have used the HHRAP model for simulating TFA fate and transport. This has been briefly described between Lines 253-260 of the (submitted) manuscript under the methods section.

Figure 6: The GEOS-Chem gridded TFA deposition over the selected river drainage basins often bridge both the drainage basin and the surrounding area. How are these deposition rates accounted for in the sections where the gridded deposition has both regions contained within it?

**Response**: As described in the comment response for lines 232-235, to account for a grid cell overlapping a basin and an area outside the subbasin, a weighted average calculation was used to provide a more appropriate estimation of the TFA deposition within the basin boundaries. The TFA deposition flux for each subbasin was calculated using the gridded dataset by computing a weighted average that accounts for the overlapping area of each grid cell with the subbasin.

Lines 340 – 343: Why are the authors assuming the TFA deposition fluxes do not change temporally? Does Fig. S3 and S4 not show that these do vary both temporally and spatially for dry and wet deposition, respectively.

**Response**: We agree that Figures S3 and S4 show temporal and spatial variability in deposition fluxes. However, for the purpose of estimating long-term (30-year) cumulative TFA accumulation in surface water, soil, and sediment, we assumed no inter-annual variability in annual TFA deposition fluxes, and we used annual deposition (rather than seasonal deposition) in the surface fate and transport modeling. This simplification was made to enable a first-order estimate of long-term accumulation, given the absence of multi-year model simulations. Also, it is unknown how the seasonality itself would vary over the 30-year period. For clarity, we have rewritten the concerning section in the revised manuscript as follows,

"Assuming no inter-annual variability in the annual TFA deposition fluxes (Table 3), we estimated net TFA accumulations in the surface water, soil, and sediment of the three watersheds over a 30-year period. While TFA deposition varies seasonally, the focus here is on the accumulation of TFA over the 30-year period. Also, it is unknown how the seasonality itself would vary over the 30-year period. So the surface fate and transport modeling was based on the mean annual deposition flux."

Lines 404 – 414: I find these three points hard to read and believe that they should be rewritten. Additionally, I find the language used to be quite emotive and political. Rather than pushing an agenda, results should be presented scientifically and then contextualised, but taking the assumptions and limitations of the study into account.

**Response**: We agree that those three points are hard to read. In the revised manuscript, we have revised them to the following,

"Atmospheric and watershed modeling predicts that future use of pMDIs may lead to TFA concentrations between 0.8 and 19.3 ng L$^{-1}$ in surface waters, 2.3 and 8.8 ng kg$^{-1}$ in surface soils, and 0.2 and 4.8 ng kg$^{-1}$ in surface sediments across the three studied watersheds. These variations reflect local factors such as water flow, region-specific deposition rates, pMDI usage patterns, and weather conditions. The predicted TFA levels can be put into context by comparing them with established reference values below:

    *i)*      The German EPA's precautionary threshold for TFA in drinking water is 10,000 ng L$^{-1}$ (Arp et al., 2024). The modeled TFA concentrations attributed to pMDI use are over 500 times below this limit.

    *ii)*      The REACH dossier establishes a long-term no-observed effect concentration (NOEC) for TFA in soil at 830,000 ng kg$^{-1}$ (Arp et al., 2024). The predicted soil levels from pMDI emissions are at least 90,000 times lower than concentrations known to affect plant shoot growth.

    *iii)*      Given the lowest reported concentration of TFA in Rhine surface water—400 ng L$^{-1}$ (Sturm et al., 2023)—the results indicate that prospective NGP emissions from pMDI use would contribute less than 1% of the total TFA in the Rhine.

Overall, these results indicate that, even if HFO-1234ze(E) were to become the sole medical propellant in future pMDIs, its continuous atmospheric emissions would result in only very low quantities of TFA in surface water and surface soil—several orders of magnitude below concentrations associated with

human or crop toxicity. In addition, the near-zero global warming potential of HFO-1234ze(E) supports its consideration as a potential long-term alternative to existing medical propellants."

Lines 416 – 420: As for the previous point, this whole paragraph is pushing an agenda, whereas it would be better to outline that these results are from sales figures from a single pharmaceutical company and that emissions of HFO-1234ze(E) are not limited to this one source or company. Instead, it contributes to total emissions.

**Response**: Note that the pMDI sales data that we have used for HFO-1234ze emission calculations is from all manufacturers, i.e., not tied to one company. We concede that in ideal scenarios we should have considered scaled the pMDI emission rate to match the predicted increase in COPD/Asthma population. We have mentioned this drawback of our study in the new "Limitations of the Study" section. In the revised manuscript, we have also revised the concerning paragraph, which states the suitability of HFO-1234ze(E) as a long-term medical propellant, to have a neutral tone.

**Technical corrections**

Line 13: HFO is not defined prior to its first use.

**Response**: The full name of HFO-1234ze(E) has been included in the Abstract of the revised manuscript. As ACP abstracts are limited to 250 words, in the revised manuscript, we have now edited the entire abstract to accommodate this change (see below).

"**Abstract.** Trans-1,3,3,3-tetrafluoroprop-1-ene (the E-isomer), written as $E-CF_3CH=CHF$ or HFO-1234ze(E), is a hydrofluoroolefin (HFO) compound developed for use in pressurised metered-dose inhalers (pMDIs) as a next-generation medical propellant (NGP). This compound has a '$-CF_3$' moiety, which makes formation of trifluoroacetic acid (TFA) possible in the atmosphere. To quantify the contribution of these prospective pMDIs in forming TFA, we performed an extensive study using a global atmospheric model coupled with detailed watershed modelling. Our approach incorporated the master chemical mechanism of HFO-1234ze(E), considering all known atmospheric pathways for TFA formation, and assumed pMDI usage as the sole emission source of HFO-1234ze(E). Based on global pMDI volume sales data, we estimate HFO-1234ze(E) emission rates to be 4.736 Gg $yr^{-1}$. Although emissions are higher in northern-temperate regions, our model predicts that the highest TFA deposition rates occur in the tropics, which is likely due to more intensive photolysis of trifluoroacetic aldehyde in temperate zones, which favours non-TFA products and/or transport of TFA into the tropical zone from nearby regions. Using predicted TFA deposition around the Hudson, Cauvery, and Rhine rivers, we applied a fate-and-transport model to estimate TFA concentrations in surface water, soil, and sediments within these watersheds. Our watershed modelling results indicate that surface water TFA concentrations would vary between 0.8–19.3 ng $L^{-1}$, corresponding to a margin of exposure exceeding 500-fold for drinking water. These findings suggest that TFA formation resulting from pMDI-related HFO-1234ze(E) emissions is minimal and does not present a risk to human health or the environment."

Line 16: What is environmental TFA? Suggest removing "environmental" as it is unnecessary.

**Response**: Corrected.

Line 36: Please use a more up-to-date GWP100 value for HFC-134a. The Annex to the WMO Scientific Assessment of Ozone Depletion 2022 has a more recent value of 1,470.

**Response**: Updated the GWP100 of HFC-134a to 1,470.

Line 41: HFO-1234ze(E) wasn't purely developed by the pharmaceutical industry as a propellant and is currently being used as a refrigerant and foam blowing agent. Emissions are already observed globally by the AGAGE network (see https://doi.org/10.1021/es505123x). Please reword this sentence.

**Response**: The Reviewer is correct that HFO-1234ze(E) was not developed exclusively by the pharmaceutical industry. Its development involved a partnership with Honeywell, which developed and patented the medical-grade version of HFO-1234ze(E). It is important to note that there are several differences between the medical-grade and industrial-grade versions of HFO-1234ze(E), which is used as a foam-blowing agent. The most significant being that the medical-grade version must be approved by regulatory agencies such as the FDA for use in human inhalation.

Lines 49 – 52: As far as I am aware, the PFAS regulation was passed in the EU and should be coming into law soon. Please update this sentence to reflect this.

**Response**: This is not accurate, particularly regarding the medical sector. Discussions on this sector only began this month, as indicated in the following document: link. Furthermore, we have not yet received an opinion from ECHA, which is required before the Commission can draft any legislative proposal.

Lines 62 – 65: Please outline the other TFAA degradation pathways that have been suggested in the literature. From the literature cited, it looks like the authors are mostly suggesting the photolysis of TFAA to produce HFC-23 but it is not entirely clear. There are pathways of TFAA oxidation through hydrolysis that can produce TFA – see Figure 8 in Berhringer et al (2021; https://www.umweltbundesamt.de/sites/default/files/medien/5750/publikationen/2021-05-06_texte_73-2021_persistent_degradation_products.pdf) for more information.

**Response**: Figure 1 shows the MCM of HFO-1234ze(E), with TFAA hydrolysis represented by dotted lines. As this is an important pathway, we are currently conducting experiments to determine the fate of the TFAA hydrate. However, since the rate constants for this pathway are not well established, it has not been included in our mechanism. A brief note regarding this is also provided in the figure caption.

Line 65 – 66: The assertion that no prior work has explored TFA production from HFO-1234ze(E) is slightly misleading. The authors have produced a similar study using a boxmodel with chemical degradation mechanisms included (see https://doi.org/10.3389/fenvs.2023.1297920).

**Response**: In the revised manuscript, we have clarified that except box-modeling of HFO-1234ze(E), no other work has addressed formation of TFA from HFO-1234ze(E)

Lines 75 – 81: Please include that GEOS-Chem is a Eularian model.

**Response**: While the phrase "global three-dimensional chemical transport model" used on line 75 inherently refers to a Eulerian modeling framework, we appreciate the opportunity to improve clarity and have updated the manuscript to explicitly refer to GEOS-Chem as a "global three-dimensional Eulerian chemical transport model."

Line 119: If stating the authors name in the text, it does not then need to be included in the citation in parenthesis – "… in GEOS-Chem by Mao et al. (2010) for the …". See the ACP house style for more information (https://www.atmospheric-chemistry-andphysics.net/submission.html#references)

**Response**: Thank you for the feedback. In the revised manuscript, we have updated in-line references to match ACP house style.

Lines 139 - 140: This sentence is not complete.

**Response**: Thank you for the feedback. We have fixed this formatting issue.

Line 375: Replace "Alphine" with "Alpine".

**Response**: Corrected.

Line 389: state what the GWP100 For HFO-1234ze(E) is.

**Response**: In the revised manuscript, the direct GWP100 of HFO-1234ze(E) has been mentioned in the introduction.

Fig. S6: Correlation values are usually stated as an R value in statistics, whereas pvalues are the statistical significance of each test. Both should be reported.

**Response**: In Fig. S6, we have computed Spearman's correlation, which is denote by the Greek symbol rho ($\rho$). In the revised manuscript, we have explicitly stated to make the reader aware because mentioning values for both rho ($\rho$) and $p$ may be even more confusing for the reader.

**References**

1. Andersen, M.P.S. and Nielsen, O.J.: Tropospheric photolysis of $CF_3CHO$. Atmospheric Environment, 272, 118935, 2022.
2. Andersen, M.P.S., Madronich, S., Ohide, J.M., Frausig, M., and Nielsen, O.J.: Photolysis of $CF_3CHO$ at 254 nm and potential contribution to the atmospheric abundance of HFC-23. Atmospheric Environment, 314, 120087, 2023.
3. Arp, H. P. H., Gredelj, A., Glüge, J., Scheringer, M., and Cousins, I. T.: The global threat from the irreversible accumulation of trifluoroacetic acid (TFA), Environmental Science & Technology, 58, 19925-19935, 2024.
4. Behringer, D., Heydel, F., and Gschrey, B.: Persistent degradation products of halogenated refrigerants and blowing agents in the environment. Type, environmental concentrations, and fate with particular regard to new halogenated substitutes with low global warming potential, Final report (No. UBA-FB--000452/ENG), Umweltbundesamt (UBA), 2021.
5. Chiappero, M.S., Malanca, F.E., Argüello, G.A., Wooldridge, S.T., Hurley, M.D., Ball, J.C., Wallington, T.J., Waterland, R.L., and Buck, R.C.: Atmospheric chemistry of perfluoroaldehydes (C x $F_2$ x+ 1CHO) and fluorotelomer aldehydes (C x $F_2$ x+ 1CH2CHO): quantification of the important role of photolysis. The Journal of Physical Chemistry A, 110, 11944-11953, 2006.
6. Javadi, M. S., Søndergaard, R., Nielsen, O. J., Hurley, M., and Wallington, T.: Atmospheric chemistry of trans-CF 3 CH= CHF: products and mechanisms of hydroxyl radical and chlorine atom initiated oxidation, Atmospheric Chemistry and Physics, 8, 3141-3147, 2008.
7. Jenkin, M. E., Saunders, S. M., and Pilling, M. J.: The tropospheric degradation of volatile organic compounds: a protocol for mechanism development, Atmospheric Environment, 31, 81-104, 1997
8. Leather, K. E., McGillen, M. R., Ghalaieny, M., Shallcross, D. E., and Percival, C. J.: Temperature-dependent kinetics for the ozonolysis of selected chlorinated alkenes in the gas phase, International Journal of Chemical Kinetics, 43, 120-129, 2011.
9. McGillen, M. R., Fried, Z. T., Khan, M. A. H., Kuwata, K. T., Martin, C. M., O'doherty, S., Pecere, F., Shallcross, D. E., Stanley, K. M., and Zhang, K.: Ozonolysis can produce long-lived greenhouse gases from commercial refrigerants, Proceedings of the National Academy of Sciences, 120, e2312714120, 2023.
10. Sturm, S., Freeling, F., Bauer, F., Vollmer, T., aus der Beek, T., and Karges, U.: Trifluoroacetate (TFA): Laying the foundations for effective mitigation – Spatial analysis of the input pathways into the water cycle, 2023.
11. Thomson, J.D., Campbell, J.S., Edwards, E.B., Medcraft, C., Nauta, K., Pérez-Peña, M.P., Fisher, J.A., Osborn, D.L., Kable, S.H., and Hansen, C.S.: Fluoroform ($CHF_3$) Production from $CF_3CHO$ Photolysis and Implications for the Decomposition of Hydrofluoroolefins and Hydrochlorofluoroolefins in the Atmosphere, Journal of the American Chemical Society, 147, 33-38, 2024.
12. Wang, Z., Wang, Y., Li, J., Henne, S., Zhang, B., Hu, J., and Zhang, J.: Impacts of the degradation of 2, 3, 3, 3-tetrafluoropropene into trifluoroacetic acid from its application in automobile air conditioners in China, the United States, and Europe, Environmental science & technology, 52, 2819-2826, 2018.

---

## Author Response (AR1)

**Atmospheric and watershed modelling of HFO-1234ze(E) emissions from prospective pressurized metered-dose inhalers usage** by Shivendra G. Tewari et al.

MS Preprint egusphere-2025-1031

The manuscript by Shivendra G. Tewari presents a study of the atmospheric degradation of HFO-1234ze(E) and hydrological fate of one of its degradation products, the environmentally persistent trifluoroacetic acid (TFA). The study is specific for the emissions of HFO-1234ze(E), used as a propellant in metered-dose inhalers, and does not discuss emissions of other use cases. Although, the applied atmospheric and hydrological modelling appear appropriate in the first place, there is very little information on validation of the model results in terms of discussion of previous simulations and observations. In addition, the focus on three selected watersheds for the hydrological part of the paper seems very arbitrary and not suited to provide global worst-case scenarios. Hence, the main conclusion of the study, that the use of HFO-1234ze(E) for inhalers will not pose a risk for TFA in drinking water, may not be valid everywhere on the globe and should be revisited. Although, the manuscript is well-structured and written and only a few additional clarifications are required in this respect, addressing the main concerns may require major revisions of the manuscript before publication.

**Response:** We thank the Reviewer for their feedback. In the revised manuscript, we have addressed all their concerns. Below, we provide our point-by-point responses to each issue raised.

**Major comments**

**Generalisation of results**: Although, it is explicitly said that results are derived for three selected watersheds, the final sentence of the abstract and the final paragraph of the conclusions seems to imply that the use of HFO-1234ze(E) in inhalers does to pose a threat globally. The selection of these watersheds is very arbitrary (L227/228). "prominent" and "large population" seem to be the only criteria. However, for all three continents one can very easily come up with watersheds that have a larger population. Figure 4 shows that there are also word regions where we expect larger TFA deposition. However, in the end it is not the deposition flux but the rainwater concentration that will determine surface water concentrations. Hence, I would have expected that a selection of watersheds would focus on those for which largest TFA concentrations in precipitation are predicted. Previous studies have shown that TFA concentrations will be enhanced in more arid regions and, hence, a look at river systems in such areas would be more helpful than considering the 3 current watersheds, from which we cannot conclude that TFA levels will stay within the safety margins globally.

**Response:**
In this study, our objective was to investigate TFA levels in surface waters within three representative watersheds. The Rhine watershed in Europe was selected primarily because it offers a rich body of TFA-related literature, enabling comparative analysis and contextualization of our findings. The US was selected because it has the highest pMDI sales/usage globally. For the third watershed, India was chosen as it represents the third-largest pMDI market globally—after the US and UK. Since our study already included a European watershed (the Rhine), we opted for India instead of the UK to maximize geographical diversity.

Within the US, several watersheds were considered, but the Hudson one was selected for its proximity to New York City, which has the country's highest population density and, thus, presumably, has the highest pMDI usage, i.e., propellant emissions. The selection of the Cauvery watershed in India was guided by the availability of watershed modelling parameters and the fact that it represents a source of drinking water to a very large population centre, i.e., Bengaluru. In the revised manuscript, we have elaborated on this selection rationale to provide greater transparency.

We appreciate the Reviewer's observation regarding the interplay between atmospheric TFA mass and precipitation volume in determining surface water concentrations. Our choice of watersheds prioritized regions with high pMDI emissions, which should reasonably correspond to areas of higher atmospheric TFA mass. While we acknowledge that arid regions may experience higher surface water concentrations due to lower precipitation volumes and increased relevance of dry deposition—an aspect illustrated in Figure 4, which shows more significant dry deposition in such areas—pMDI sales and propellant emissions are comparatively lower in these arid regions. Note that even though arid regions may experience higher TFA rainwater concentrations, the net TFA deposition in these regions will be lower due to limited precipitation (Vet et al., 2014). Given the scope and resource constraints of the present study, it was not feasible to include a fourth watershed focused on an arid environment for detailed fate-and-transport analysis.

**Isolated view on the TFA budget**: There is one more danger in the presentation of isolated results of TFA from a single precursor and use case. Obviously, it is the sum of contributions from all precursors that determines environmental TFA levels. If ten studies like the current for specific use cases all come to the conclusion that individually there is no problem, the sum may still present an environmental problem. It is mentioned that compared to current levels of TFA in precipitation in Germany, TFA from degradation of HFO-1234ze(E) in pMDIs would add less than 1 %. Again, this may not be true globally. I would suggest to put the current use case more into the perspective of the global TFA budget (as much as this is known, for example see Madronich et al. 2023).

**Response**: We thank the Reviewer for highlighting this concern and indeed it is important to consider all possible precursors of TFA in the environment and that was the reason for comparing existing TFA surface water concentration in Rhine (Germany), which presumably encompass all sources of TFA, with our estimates of surface water TFA due to pMDI-usage based propellant in that region. It is indeed possible that relative contribution of propellant-based TFA in surface water would be different in different regions; therefore, comparison with global TFA budget is the best possible alternative. As per Madronich et al 2023, HFC-134a and HFO-1234yf are the two main sources of TFA in the environment. Their combined annual production rate of TFA is $0.04 – 0.06$ Tg yr$^{-1}$. Considering our estimates of annual pMDI-associated HFO-1234ze(E) emissions and theoretical TFA yield, the global total TFA deposition due to future pMDI usage would be ~0.0002 Tg yr$^{-1}$, i.e., $4.736 \times 0.04$, which suggests that propellant emissions based TFA production represents less than 0.5% of the annual global TFA in the environment estimated by Madronich et al. 2023. In the revised manuscript, we have added this new information as well.

**Emission scenario**: It is assumed that future emissions from pMDIs will follow the same global usage as taken from current pMDI sales (L271). Figure 3 reveals that some world regions are underrepresented with this assumption. Especially emissions in densely populated China seem to be unrealistically low. Is this because sales data from China is potentially incomplete? Even if it is complete, would one not expect that access to pMDIs will increase in China in the future? In general, it would be better to work with projected consumption numbers then with present day values. Since Southeast Asia also seems to be an area of intense TFA deposition (Figure 4b), realistic Chinese emissions seem to be critical for a fair assessment of future TFA levels in this region. Furthermore, instead of using NO as a proxy for the spatial distribution of HFO emissions, it seems more appropriate to use population density directly. NO distributions may be skewed by individual point sources like power plants.

**Response**: We thank the Reviewer for this prudent observation. Note that the lower propellant emissions in China is because the IQVIA pMDI sales data for that region is only available from Hospitals. With the revised manuscript, we have provided out estimates of monthly propellant emission rates for the top 51 countries as supplement data; therein, we have annotated that lower emissions in China are due to lack of data availability.

We agree that there are limitations to this study, such as assuming pMDI sales do not increase over time. Without hard evidence, it is difficult predict what world regions will see the most increase in pMDI sales. Moreover, we cannot predict weather patterns for the next thirty years which should also affect TFA formation and deposition globally. Therefore, for relative simplicity, we used region-specific annual TFA deposition flux for calculating net accumulation of TFA in surface water, soil, and sediment over time. In the revised manuscript, we have addressed this limitation under a new "Limitations of the Study" section.

As specified in the manuscript, the NO emissions used as a spatial proxy for HFO-1234ze(E) emissions are only from the residential and commercial sectors of the CEDS inventory. This was done intentionally, as these sectors are representative of population density and human activity and are therefore more likely to correlate with pMDI usage. Emissions from unrelated sources—such as biogenic, aircraft, or maritime—were not included in this proxy

**Minor comment**

**Section 2.1-2.3:** There are several questions concerning the setup of GEOS-Chem that need clarification. What is the name/version of the utilised chemistry scheme? How many compounds are treated? How are the NMVOC emissions mapped onto model species? Are there any validation results of GEOS-Chem for the classical air pollutants (O3, NOx, …)? Done as part of this study or published elsewhere for the same setup of the model. These would be helpful to understand if simulated OH levels and, hence, HFO reaction rates are realistic.

**Response**: We used GEOS-Chem version 14.2.2 with the standard full-chemistry simulation, as described in lines 75–76 of the submitted manuscript and referenced accordingly. This version includes 324 chemical species, which comprises the standard species set plus 15 additional species added for HFO chemistry in this study.

Non-methane volatile organic compound (NMVOC) emissions are pre-speciated in the emission inventories (e.g., HTAP, RETRO, EDGAR, NEI) used in GEOS-Chem. That is, emissions are mapped directly to individual model species (e.g., ISOP, ACET, ALK4) at the point of input. There is no in-model speciation of total NMVOC; instead, species are defined based on inventory-provided breakdowns.

Regarding model evaluation, GEOS-Chem has been extensively validated in previous studies for classical air pollutants such as $O_3$, $NO_x$, CO, and related species. Some key references include:

- Hu et al. (2017): Evaluated the global ozone budget and tropospheric chemistry in v10-01 using satellite (OMI), aircraft (IAGOS), and ozonesonde data.
  https://doi.org/10.1016/j.atmosenv.2017.08.036

- Wang et al. (2022): Assessed ozone trends and radiative impacts using v13.3.1 with multiple long-term observational datasets (IAGOS, ozonesondes).
  https://doi.org/10.5194/acp-22-13753-2022

- Lin et al. (2024): Compared GEOS-Chem v14.1.1 with CAM-Chem in CESM2, focusing on oxidant chemistry.
  https://doi.org/10.5194/acp-24-8607-2024

- David et al. (2019): Evaluated tropospheric ozone over the Indian subcontinent using v10-01.
  https://doi.org/10.1016/j.atmosenv.2019.117039

Additionally, the GEOS-Chem team conducts comprehensive benchmark comparisons for each major model version to ensure scientific consistency and detect any significant changes in model behaviour. These

benchmarks track changes in model outputs such as ozone, $NO_x$, CO, and VOCs across versions to ensure internal consistency, but they are not designed as full evaluations against observations. For example, a benchmark was performed for version 14.2.0, but not for minor updates such as 14.2.1, 14.2.2, or 14.2.3, which typically involve bug fixes or minor code changes. Benchmark results for version 14.2.0 are available at: https://wiki.seas.harvard.edu/geos-chem/index.php/GEOS-Chem_14.2.0

Separate evaluation studies—many published—compare GEOS-Chem outputs against surface, aircraft, and satellite observations, providing confidence that key aspects of the model's chemistry and transport are scientifically robust.

**L153f:** How valid is the use of absorption cross sections of CH3C(O)CHO for CF3C(O)C(O)F? Can this be corroborated from similarities from any know cross sections for other similar molecules?

**Response**: Thank you for raising this important point regarding the choice of absorption cross sections for $CF_3C(O)C(O)F$. As direct experimental data for the absorption cross section of $CF_3C(O)C(O)F$ are currently unavailable, we followed established precedents in the photochemical literature—specifically the approach outlined by Jenkin et al (2019). In such cases, it is common practice to use cross sections from structurally analogous compounds as surrogates.

$CH_3C(O)CHO$ was selected as a substitute based on its analogous carbonyl functionality and overall structural similarity to $CF_3C(O)C(O)F$, particularly in terms of the presence of conjugated carbonyl groups that strongly influence UV absorption behaviour. The selection is further supported by the fact that photolysis in these compounds typically occurs via the $n{\rightarrow}\pi^*$ transition in the carbonyl group, for which absorption cross sections have demonstrated comparable spectral features among simple and perfluorinated aldehydes and ketones; see, for example, Orlando and Tyndall (2012). While small differences can arise due to the electronic effects of fluorination, the absence of directly measured data justifies the use of this pragmatic surrogate.

We acknowledge that acquiring measured cross sections for $CF_3C(O)C(O)F$ would improve accuracy and recommend this as a focus for future work. In the meantime, our choice is in line with methods used in previous atmospheric modelling studies for compounds lacking explicit spectroscopic data.

**Figure1 and section 2.3:** The figure seems to indicate an alternative path for trifluoroacetic aldehyde (TFAA) to TFA through hydration. This path is not discussed in the text. Is this based on the diol mechanism suggested by Franco et al. (2021) for other aldehydes? Please comment if it was included in the chemistry scheme and if it showed any relevance.

**Response:** The reviewer is correct that the dotted path from TFAA to TFA in Figure 1 represents the in-cloud hydration of TFAA. To clarify, TFAA hydration and TFAA diol formation refer to the same process: when trifluoroacetic aldehyde ($CF_3CHO$) reacts with water, it forms the corresponding geminal (1,1-) diol, $CF_3CH(OH)_2$. This hydrated intermediate may then partition back to the gas phase, where it is thought to react with OH radicals, potentially contributing to TFA formation. The possible involvement of gem-diol formation for TFAA in atmospheric waters, as discussed in Franco et al. (2021), remains a proposed mechanism; to our knowledge, no direct experimental evidence yet confirms TFA formation proceeding specifically through this hydrated intermediate. Critical information is still lacking, such as reliable values for the Henry's Law coefficient of TFAA, hydration/dehydration kinetics in the aqueous phase, the rate coefficient for the OH + $CF_3CH(OH)_2$ reaction, and the gas-phase dehydration kinetics of the diol. Recognizing its potential importance, we are currently conducting experiments to better understand the fate and reactivity of the TFAA hydrate. A note to this effect has also been added to the Figure 1 caption in the revised manuscript.

**Section 2.4**: Does the model take evapotranspiration in the watershed into account? Should it be considered in warmer climates.

**Response**: We can confirm that the model does account for evapotranspiration, with region-specific values applied in our simulations for each of the three studied watersheds. These values are detailed in the Supplementary Information. We agree that incorporating more spatially refined values would be particularly valuable for studies in warmer climates where evapotranspiration plays a critical role. This will be an important focus for future work, aiming to improve the resolution and accuracy of our simulations.

**Figure 4 and Table 3**: Earlier model studies on HFO degradation and TFA deposition all came to the conclusion that wet deposition dominates over dry deposition. This was the case for HFO-1234yf, where the formation of TFA should be fast (e.g., Luecken et al., 2010; Henne et al., 2012, Wang et al., 2018), but also for HFO-1233zd(E) (Sulbaek Andersen et al., 2018) for which TFA formation also proceeds through TFAA. Please comment, why and how this could be different in the present case. This also questions the statement made on line 298 concerning 'akin' deposition fluxes as in (Sulbaek Andersen et al., 2018), which then seems oversimplified.

**Response**: While earlier studies (e.g., Luecken et al., 2010; Henne et al., 2012; Wang et al., 2018; Sulbaek Andersen et al., 2018) concluded that wet deposition dominates TFA removal, our results suggest that dry deposition may play a more significant role under specific tropical and subtropical conditions. This difference likely stems from the chemical formation pathway (via TFAA) and regional meteorology that affects deposition processes.

1. Enhanced tropical photochemistry

In the tropics, intense solar radiation and high temperatures accelerate:

- $CF_3C(O)O_2$ formation via OH oxidation of $CF_3CHO$:

$$CF_3CHO + OH\ (+O_2) \rightarrow CF_3C(O)O_2$$

- TFA production through the reaction $CF_3C(O)O_2 + HO_2 \rightarrow CF_3C(O)OH + O_3$, driven by elevated $HO_2$ concentrations.

These reactions lead to elevated TFA levels in the lower troposphere.

2. Suppressed wet deposition efficiency

Despite frequent rainfall, tropical meteorological conditions reduce the effectiveness of wet scavenging:

- Shallow convection: Rain is often produced by clouds that are limited in vertical extent and do not efficiently mix with near-surface air, where much of the TFA resides.

- High evaporation rates: In hot, humid conditions, raindrops can evaporate before reaching the ground, re-releasing dissolved TFA back into the atmosphere.

- Short cloud processing times: Warm, fast-evolving clouds allow little time for gases like TFA to dissolve into droplets, reducing uptake efficiency.

3. Favourable conditions for dry deposition

With wet removal suppressed and chemical degradation slow, more TFA remains near the surface. In subtropical regions (descending branches of the Hadley cell), conditions such as low rainfall, dry soils, and sparse vegetation promote efficient dry deposition, making it the dominant sink in these areas.

In summary, our study highlights that, under specific tropical and subtropical conditions, dry deposition can rival or exceed wet deposition as a removal pathway for TFA, a finding not emphasized in previous work. In the revised manuscript, we have briefly discussed this distinction.

**Figure 4:** In order to assess the global impact of TFA deposition, it would be beneficial to add another figure that shows average rainwater concentrations of TFA. Since these are usually strongly enhanced during the summer months, I would suggest to show these with three panels: overall average, summer average, winter average. Derived concentrations could be used in the discussion against currently observed TFA in precipitation (see below).

**Response:** Thank you very much for the helpful suggestion. Because summer and winter months vary across the globe in response, we have added a new Figure 8 in the manuscript showing overall average and a supplement figure (Fig. S14) presenting monthly modelled mean TFA rainwater concentrations, calculated as the ratio of monthly modelled wet deposition flux to monthly precipitation, which is taken from MERRA-2 reanalysis that is an assimilated product. To address your further point, we now compare our average modelled rainwater concentrations with values reported in the scientific literature for corresponding continents:

Figure 8 reveals that, across all continents, model-predicted TFA rainwater concentrations are substantially lower than those typically reported in the literature. Specifically, our modelled values range from about 0.005 to 0.040 μg/L (5–40 ng/L) globally, with most regions—including North America, Europe, and Asia—falling at the lower end of this range. In contrast, literature values compiled from large-scale precipitation sampling report TFA concentrations that are often several times higher. For example, measured concentrations in North America frequently fall between 10 and 340 ng/L (Solomon et al., 2016), in Europe (including Germany) medians range from 69 to 350 ng/L (Freeling et al., 2020), and in Asia (China and Japan) values as high as 60–550 ng/L have been reported (Wang et al., 2014; Yamanaka et al., 2012). Note that even though arid regions, such as North Africa and the Middle East, have higher TFA rainwater concentrations, the net TFA deposition in these regions will be lower due to low precipitation (Vet et al., 2014). We have provided these comparisons in the revised manuscript (Figure 8).

**Section 3.1:** What was the total global TFA deposition flux? How does it compare to the HFO emissions and what can be concluded in terms of TFA yields for this compound?

**Response:** We thank the Reviewer for raising this important point. The primary goal of this manuscript was to evaluate TFA concentrations in surface water, soil, and sediment resulting from prospective pMDI usage and the associated release of HFO-1234ze(E) emissions into the atmosphere. While our model provides estimates of total global TFA deposition flux, calculating precise TFA yields—even with a single environmental precursor—requires rigorous integration of emission inventories, atmospheric processes, and chemical pathways. This level of quantitative analysis, directly linking emissions to deposition, was beyond the scope and objectives of this study.

Recognizing the importance of this issue, we note that our group is actively pursuing dedicated research to determine global TFA yields with the necessary precision. To maintain the clarity and integrity of our current work and ongoing studies, we have therefore not included a direct comparison of propellant emissions with TFA deposition fluxes. These critical aspects will be addressed in forthcoming publications.

**Figure 5 and L325f**: To me it remains unclear how the conclusions can be reached from what is presented in the figure. Is it only the spatial correlation between the two quantities? But then TFA concentrations will strongly depend on the removal not just the production pathway. Furthermore, how does the correlation look for other intermediates? This requires additional explanation.

**Response**: We thank the Reviewer for highlighting this important point and appreciate the opportunity to elaborate. To directly address the concern: while it is generally true that the steady-state concentration of any atmospheric species is governed by both its production and removal, in the case of gas-phase TFA, our analysis reveals a distinct scenario.

The key detail is that gas-phase TFA is removed from the atmosphere almost exclusively through a single OH-driven, temperature-independent oxidation process (Reaction 67 in Table S1). The rate of this removal pathway is significantly lower—by about an order of magnitude—than the rates of the major TFA formation reactions, including the OH-dependent reaction that forms TFA. As a result, the atmospheric burden and spatial patterns of gas-phase TFA are primarily determined by spatial patterns of chemical species that form TFA, rather than remove it. [Note that any OH-dependent changes in the removal rates of TFA would be negated by the corresponding OH-dependent reaction that forms TFA.]

We demonstrate this by showing that the spatial distribution of TFA closely matches that of the ratio $([HO_2] \times [CF_3C(O)O_2])/[OH]$, which includes species that form and remove TFA in the atmosphere. The numerator reflects the major formation processes, while the denominator includes the OH species responsible for both TFA formation and removal. Thus, the observed spatial correlation in the figure is not simply a reflection of two unrelated quantities but encapsulates the dominant chemistry controlling TFA's presence in the atmosphere.

We derived this spatial correlation of the ratio by performing a detailed analysis of the chemical reactions that form and remove TFA from the atmosphere. With the supplement of the revised manuscript, we now provide the entire process which shows atmospheric concentration of other crucial intermediates. This new section (Section S2) demonstrates that the spatial patterns observed for TFA are distinct and are not mirrored by other intermediate species in the same way. This further supports our mechanistic interpretation.

We have clarified these points in the revised manuscript, making explicit how the data and analysis support our conclusions regarding TFA's atmospheric behaviour.

**Figure 6**: The plot reveals another potential shortcoming in the assessment of maximal TFA concentrations. The global chemistry simulations were performed at relatively coarse resolution. However, precipitation and hence TFA deposition often varies at much smaller scales as can be covered by the global chemistry model. As a consequence actual TFA inputs into individual watersheds may largely differ from the grid cell average of GEOS-Chem. Please add a note of caution and discuss the possible implications.

**Response**: We thank the Reviewer for raising this concern. The 2° x 2.5° resolution of the TFA deposition data does not fully capture granularity, specifically in smaller subbasins, but this resolution is necessary considering the global scale of the modeling and is typical of such modeling. In the revised manuscript, we have added the limitations of using such a resolution in the new "Limitations of the Study" section.

**L340:** There seems to be another important simplification for the hydrological modelling which needs to be addressed. It is well know that TFA inputs from the atmosphere have a strong seasonal cycle in midlatitudes (both observed and simulated rainwater concentrations show this). In order to assess maximum concentrations in the watersheds it therefore seems very important to consider the seasonality in the inputs and see how this variability propagates through different strata.

**Response**: We appreciate the Reviewer's comment regarding the pronounced seasonal cycle in atmospheric TFA deposition, as also reflected in Figures S3 and S4, which present the temporal and spatial variability in deposition fluxes. For this study, our primary goal was to estimate long-term (30-year) cumulative TFA accumulation in surface water, soil, and sediment. To provide a first-order estimate suitable for this timescale, we assumed no inter-annual variability in annual TFA deposition, using the mean annual deposition rates as input to the surface fate and transport modelling. This approach was necessitated both by the scope of available deposition data and the absence of multi-decade, high-resolution model simulations, as well as by the uncertainty in how deposition seasonality itself might change over such an extended period.

We acknowledge that this simplification may underestimate short-term or peak TFA concentrations in particular strata resulting from seasonal maxima in deposition. We have revised the relevant section in the manuscript to clarify this point and to guide interpretation of the results:

"Assuming no inter-annual variability in the annual TFA deposition fluxes (Table 3), we estimated net TFA accumulations in the surface water, soil, and sediment of the three watersheds over a 30-year period. While TFA deposition varies seasonally, the focus here is on the accumulation of TFA over the 30-year period. Also, it is unknown how the seasonality itself would vary over the 30-year period. So the surface fate and transport modeling was based on the mean annual deposition flux."

We trust this revision clarifies the rationale for our modelling approach and its limitations.

**L353f and L362:** Both statements seem to suggest that a large fraction of TFA will accumulate in the soil. To me it is not clear on which time scales you are discussing this. In steady state (which apparently is reached quickly), input from the atmosphere should be equal to outflow to the ocean. Or is what you call deep soil a open boundary for the model as well? Furthermore, this strong flux to soil seems to contradict the statement in the footnote of Table 1: " Adsortion/desorption tests results show that TFA is poorly adsorbed to the soil and is considered as a mobile organic compound in the majority of soils investigated." Please clarify.

**Response**: We appreciate your helpful comment. To clarify, TFA is considered highly mobile in soil due to its low octanol-water partition coefficient (Kow), which indicates a strong preference for remaining in the aqueous phase rather than adsorbing to soil organic matter. Adsorption/desorption tests of TFA confirm that TFA is poorly retained by most soils, consistent with its classification as a mobile organic compound. Because of this high mobility, TFA does not accumulate in the soil over long timescales. Rather, it tends to leach downward from surface soil into subsurface layers and potentially into groundwater. The apparent accumulation in "soil" discussed in our manuscript refers to transient retention in the soil column before leaching, rather than long-term storage. In this context, "deep soil" serves as a transitional boundary in the mass balance model rather than an open system boundary or final sink. Importantly, our model does not explicitly simulate transport through the full vadose zone or groundwater systems. Instead, the mass flux from surface soil to deeper compartments is estimated through mass balance. Our primary focus is on the surface water compartment and the timescale on which TFA reaches steady state in surface waters. We have revised the text to clarify this in the discussion of mass allocation analysis to avoid confusion regarding the contradiction with TFA's known mobility in soil.

**L378**: For the Rhine catchment a more direct comparison between simulated atmospheric inputs and measured TFA in precipitation could be done (see Freeling et al., 2020).

**Response:** We thank the Reviewer for highlighting the opportunity to compare our simulated atmospheric TFA inputs with observed precipitation data, specifically those published by Freeling et al. (2020). We acknowledge that Freeling et al. provides an important dataset for TFA concentrations in precipitation (rain and snow) in the Rhine catchment, which is highly relevant for validating atmospheric deposition models. However, a validation

using observed rainwater TFA concentrations from Freeling et al 2020 would not be appropriate here because we are simulating TFA formation due to prospective pMDI usage only. Also, the focus of our study is on quantifying TFA concentrations in surface waters, rather than simulating or directly analysing atmospheric deposition fluxes or precipitation concentrations. Consequently, our direct modelling outputs, i.e., surface water, are not comparable to the precipitation measurements reported in Freeling et al. (2020) but rather to direct measurements from Sturm et al. (2023), which specifically reports TFA concentrations in Rhine surface waters. For completeness, in the revised manuscript, we have now calculated global TFA rainwater concentrations (Figure 8 and Section S3) and performed qualitative comparison of those results with that from Freeling et al. 2020 and other relevant TFA rainwater concentrations globally.

**L407f:** Why not discuss with the often quoted NOEC for the most sensitive freshwater algae, which is 120'000 ng/L? In addition, as TFA cannot be removed from drinking water at large scale, the discussion of an additional threshold much higher than the one suggested for drinking water seems a bit artificial.

**Response**: We thank the Reviewer for their thoughtful suggestions regarding the discussion of TFA concentration thresholds. In our manuscript, we used the German drinking water threshold of 10,000 ng/L as the most conservative regulatory guideline available at the time of our analysis. We acknowledge, however, that the Netherlands recently established a more stringent threshold for TFA in drinking water of 2,200 ng/L.

Regarding ecotoxicological relevance, we appreciate the Reviewer's point about the often quoted NOEC (No Observed Effect Concentration) for the most sensitive freshwater algae, which is 120,000 ng/L. We agree that including a comparison to this ecotoxicological benchmark provides important context for assessing potential ecological risks. In our study, the highest estimated TFA concentration in surface waters was ~19 ng/L. This value is orders of magnitude below both the German and Dutch drinking water thresholds, as well as the NOEC for sensitive freshwater algae.

We also agree that since TFA is highly persistent and not removable at large scale from drinking water, it is more appropriate to focus discussion on the most relevant and up-to-date regulatory limits, rather than any threshold substantially higher than the current drinking water guidelines. Following the reviewer's advice, we have revised the discussion to (i) include the recent Dutch drinking water threshold, (ii) compare our findings to the ecotoxicological NOEC for freshwater algae, and (iii) clarify that the concentrations we observed are well below all of these thresholds.

We have now revised the concluding relevant section to the following:

"Lastly, the model's predicted TFA levels can be effectively evaluated by comparison with several established reference values. The highest surface water TFA concentrations attributable to pMDI use are more than 500-fold lower than the German Environment Agency's conservative drinking water threshold of 10,000 ng L$^{-1}$ (Arp et al., 2024). Similarly, these modeled concentrations are greater than 100 times below the Netherlands' most recent drinking water guideline of 2,200 ng L$^{-1}$ (Arp et al., 2024), derived based on precautionary potency factors for PFAS. When placed in ecological context, the maximum TFA level estimated in surface water is also over 6,000 times below the frequently cited no-observed-effect concentration (NOEC) of 120,000 ng L$^{-1}$ for sensitive freshwater algae (Arp et al., 2024), indicating negligible risk to aquatic biota. For soils, predicted TFA loadings from pMDI emissions remain at least 90,000 times lower than the REACH long-term NOEC of 830,000 ng kg$^{-1}$ for plant health (Arp et al., 2024). Furthermore, given that the lowest TFA concentration empirically measured in Rhine surface water is 400 ng L$^{-1}$ (Sturm et al., 2023), prospective new emissions from pMDI use would represent less than 1% of the total TFA present in this major watershed (or catchment).

Taken together, these findings demonstrate that even if HFO-1234ze(E) were to become the sole medical propellant in future pMDIs of all manufacturers, its continual atmospheric release would result in only very low additional quantities of TFA in both surface water and soil—levels that are several orders of magnitude below thresholds for human health or ecological risk. The maximum TFA concentrations projected by this study remain much lower than all currently relevant drinking water, aquatic, and agro-environmental benchmarks. This demonstrates a substantial margin of safety, underscoring that, while TFA is environmentally persistent, its contribution from next-generation propellant use is expected to remain well within safe regulatory and ecotoxicological limits."

We thank the reviewer for helping us improve the relevance and clarity of our discussion.

**Technical comments**

**Citations in text:** Luecken et al.(Luecken et al., 2010) should be Luecken et al. (2010). Applies to all references that should not include the author.

**Response**: This and other occurrences of incorrect formats of author name-based references have been corrected in the revised manuscript.

**Andersen et al., 2018 and 2022**: Should be Sulbaek Andersen et al., 2018. Sulbaek being part of the surname not the given name.

**Response**: We have corrected the incorrect format of references to the articles by Prof. Sulbaek-Andersen in the revised manuscript throughout.

**Figure 6:** The labels for the color scale of TFA deposition rates are too small and even with zooming in the pdf cannot be deciphered. Similarly for all other labels.

**L375**: "Alpine" instead of "Alphine".

**Response**: Corrected.

**References**

Franco, B., Blumenstock, T., Cho, C., Clarisse, L., Clerbaux, C., Coheur, P. F., De Mazière, M., De Smedt, I., Dorn, H. P., Emmerichs, T., Fuchs, H., Gkatzelis, G., Griffith, D. W. T., Gromov, S., Hannigan, J. W., Hase, F., Hohaus, T., Jones, N., Kerkweg, A., Kiendler-Scharr, A., Lutsch, E., Mahieu, E., Novelli, A., Ortega, I., Paton-Walsh, C., Pommier, M., Pozzer, A., Reimer, D., Rosanka, S., Sander, R., Schneider, M., Strong, K., Tillmann, R., Van Roozendael, M., Vereecken, L., Vigouroux, C., Wahner, A., and Taraborrelli, D.: Ubiquitous atmospheric production of organic acids mediated by cloud droplets, Nature, 593, 233-237, 10.1038/s41586-021-03462-x, 2021.

Freeling, F., Behringer, D., Heydel, F., Scheurer, M., Ternes, T. A., and Nödler, K.: Trifluoroacetate in Precipitation: Deriving a Benchmark Data Set, Environ. Sci. Technol., 54, 11210-11219, 10.1021/acs.est.0c02910, 2020.

Jenkin ME, Saunders SM, Pilling MJ. The tropospheric degradation of volatile organic compounds: A protocol for mechanism development. Atmospheric Environment. 1997;31(1):81-104.

Madronich, S., Sulzberger, B., Longstreth, J. D., Schikowski, T., Andersen, M. P. S., Solomon, K. R., and Wilson, S. R.: Changes in tropospheric air quality related to the protection of stratospheric ozone in a changing climate, Photochemical & Photobiological Sciences, 22, 1129-1176, 10.1007/s43630-023-00369-6, 2023.

Orlando, J. J., & Tyndall, G. S. (2012). Laboratory studies of organic peroxy radical chemistry: an overview with emphasis on recent issues of atmospheric significance. *Chemical Society Reviews*, 41(19), 6294-6317

Vet, R., Artz, R. S., Carou, S., Shaw, M., Ro, C. U., Aas, W., ... & Reid, N. W. (2014). A global assessment of precipitation chemistry and deposition of sulfur, nitrogen, sea salt, base cations, organic acids, acidity and pH, and phosphorus. *Atmospheric Environment*, *93*, 3-100.

Wang, N., Liu, J., Wang, Y., & Yu, G. (2014). Global distribution and environmental sources of trifluoroacetate in urban ground and rain waters. *Chemosphere*, 112, 281–286. https://doi.org/10.1016/j.chemosphere.2014.03.109

Wang, Z., Wang, Y., Li, J., Henne, S., Zhang, B., Hu, J., and Zhang, J.: Impacts of the Degradation of 2,3,3,3-Tetrafluoropropene into Trifluoroacetic Acid from Its Application in Automobile Air Conditioners in China, the United States, and Europe, Environ. Sci. Technol., 52, 2819-2826, 10.1021/acs.est.7b05960, 2018.

Yamanaka, N., Shibata, Y., Uno, I., & Nanba, K. (2012). The distribution of trifluoroacetic acid in surface waters and precipitation across Japan. *Environmental Science: Processes & Impacts*, 14, 1349–1357. https://doi.org/10.1039/C2EM30231A

---

## Referee Report (RR1)

**Atmospheric and Watershed Modelling of Trifluoroacetic Acid from Oxidation of HFO-1234ze(E) Released by Prospective Pressurized Metered-Dose Inhaler Use** by Shivendra G. Tewari et al.

MS Preprint egusphere-2025-1031

This paper is a resubmission of "Atmospheric and watershed modelling of HFO-1234ze(E) emissions from prospective pressurized metered-dose inhalers usage" after public peer review on egusphere. The revised paper shows significant improvements to the original and takes on board many of the reviewers comments. Due to the content of the paper being scientifically valuable as not many studies have been published on pressurised metered dose inhaler emissions, and the improvements made, the paper merits publication in ACP after the minor corrections outlined below.

**General comments**

The addition of the final section on limitations of the study, into the production of trifluoroacetic acid (TFA) from the atmospheric oxidation of HFO-1234ze(E), outlines clearly the limitations that help to contextualise the results within the wider scientific landscape.

The inclusion of the additional atmospheric oxidations routes, notably the ozonolysis of HFO-1234ze(E) and the photolysis of trifluroacetylealdehyde (TFAA) to produce HFC-23 ($CF_3H$) greatly improves the paper, despite these routes having little impact on the overall results of the paper. The addition of this information helps to contextualise the results and mechanisms being reported. For completeness, it would be beneficial to outline what the "several other TFAA degradation pathways have been suggested in the literature" (lines 72-73 of the revised manuscript) are. This inclusion isn't imperative as it does not relate to the production of TFA; however, it demonstrates that the authors are aware of the full routes and contextualises the oxidative pathways.

The inclusion of the author derived HFO emissions based on the sales data is a welcome inclusion in the supplementary information. It is acknowledged that the inclusion of the sales data is potentially commercially sensitive, despite the data being country aggregates for all producers of pMDIs. The method used to derive these emissions is explained within section 2.2, so there is suffice information on the provenance of the data.

**Specific comments**

Whilst I understand why the Dutch drinking water threshold for TFA is cited on line 26 to contextualise the Rhine values, especially as the mouth of the Rhine is in the Netherlands, it isn't clear to me why the values for the Cauvery are compared to this threshold.

Some of the naming of references in the text seem to be incorrect and might be caused due to the use of a reference management software. These should be checked and corrected. Examples of erroneous naming in citations include: line 38 "(Organization, 2022)" instead of World Meteorological Organization, line 42 "(Union, 2024)" instead of Official Journal of the European Union.

The authors have addressed a previous comment by the reviewers on the inclusion of ozonolysis as an atmospheric oxidative pathway for HFO-1234ze(E). They have rightfully acknowledged the temperature dependence of the ozonolysis reactions, which are not included in the McGillen et al (2023) paper (lines 47 – 54 of the revised manuscript). Moreso than the temperature dependence, one would expect OH-initiated chemistry to dominate over ozonolysis due to the reaction rates. An acknowledgement of this limitation in the production in HFC-23 from HFO-1234ze(E) would be beneficial to show why the authors have not looked into the production of HFC-23 in this paper.

There is no description of how the modelled deposition data from GEOS-CHEM is superimposed onto the basin boundaries used for the watershed modelling. It would be good to include the information that was provided as a response to one of the reviewers on this topic in section 2.4

**Technical comments**

Line 56: add in "known" between 15,000 and chemicals to read "… PFAS are a group of nearly 15,000 known chemicals…".

Line 59: There are some PFAS compounds that take tens of decades to decompose. It might be better to show this rather than stating "many years".

Line 65: Add in "known" between 2,000 and chemicals to read "… presently, there are nearly 2,000 known chemicals that have…".

Line 107: Subtitle "Anthropogenic, Natural, and HFO-1234ze(E) emissions" does not make sense. Suggest altering to "Anthropogenic and Natural Trace gas, and HFO-1234ze(E) emissions".

Lines 109 – 111: Try to avoid using double parentheses, better to list species as "… reactive gases (sulfur dioxide, $SO_2$; nitrogen oxides, $NO_x$; ammonia, $NH_3$; …".

Table 1 footnotes: Adsorption spelt incorrectly on line 3 of the footnotes. Additionally, units of L/kg need to be changes to $L\ kg^{-1}$.

Line 309, replace "Gg/yr (or kiltons/year)" with $Gg\ yr^{-1}$ (or kilotonnes $yr^{-1}$).

Figure 3 units are not consistent with the rest of the paper – shown as $kg/m^2 \bullet yr$ instead of $kg\ m^{-2}\ yr^{-1}$.

Line 484: Remove "i.e. $4.736 \times 0.04 \times 10^{-3}$" as this does not add any extra information to the paper than the preceding value.

Line 493 – 494: It would be pertinent to reference the limitations of this study, or at least the following section, on the results.

Lines 506 – 510: Again, these results are based on the limitations of the study that should be acknowledged (or at least signed posted to the section on limitations) rather than stating something outright as it is misleading.

Supplemental data: Unts in the table need to be changes to Gg month$^{-1}$. Additionally, the number of significant figures need to be thought about here. Captions are needed for the two plots below the table that show propellant emissions released per month in the UK and Brazil, as well as month names.

---

## Author Response (AR2)

**Reviewer 1 comments**

Atmospheric and Watershed Modelling of Trifluoroacetic Acid from Oxidation of HFO-1234ze(E) Released by Prospective Pressurized Metered-Dose Inhaler Use by Shivendra G. Tewari et al. MS Preprint egusphere-2025-1031 This paper is a resubmission of "Atmospheric and watershed modelling of HFO1234ze(E) emissions from prospective pressurized metered-dose inhalers usage" after public peer review on egusphere. The revised paper shows significant improvements to the original and takes on board many of the reviewers comments. Due to the content of the paper being scientifically valuable as not many studies have been published on pressurised metered dose inhaler emissions, and the improvements made, the paper merits publication in ACP after the minor corrections outlined below.

General comments: The addition of the final section on limitations of the study, into the production of trifluoroacetic acid (TFA) from the atmospheric oxidation of HFO-1234ze(E), outlines clearly the limitations that help to contextualise the results within the wider scientific landscape. The inclusion of the additional atmospheric oxidations routes, notably the ozonolysis of HFO-1234ze(E) and the photolysis of trifluroacetylealdehyde (TFAA) to produce HFC-23 (CF3H) greatly improves the paper, despite these routes having little impact on the overall results of the paper. The addition of this information helps to contextualise the results and mechanisms being reported. For completeness, it would be beneficial to outline what the "several other TFAA degradation pathways have been suggested in the literature" (lines 72-73 of the revised manuscript) are. This inclusion isn't imperative as it does not relate to the production of TFA; however, it demonstrates that the authors are aware of the full routes and contextualises the oxidative pathways. The inclusion of the author derived HFO emissions based on the sales data is a welcome inclusion in the supplementary information. It is acknowledged that the inclusion of the sales data is potentially commercially sensitive, despite the data being country aggregates for all producers of pMDIs. The method used to derive these emissions is explained within section 2.2, so there is suffice information on the provenance of the data.

**Response**: We thank the Reviewer for their feedback. In the revised manuscript, we have specified the additional TFAA degradation pathways. Below, we address each comment point-by-point:

**Specific comments**: Whilst I understand why the Dutch drinking water threshold for TFA is cited on line 26 to contextualise the Rhine values, especially as the mouth of the Rhine is in the Netherlands, it isn't clear to me why the values for the Cauvery are compared to this threshold.

**Response**: Thank you for this pertinent observation. We agree that the Cauvery values cannot be directly compared to Dutch drinking water threshold. Therefore, in the revised manuscript, we now only compare the Dutch drinking water thresholds for TFA with the Rhine watershed value due to the river's mouth being in the Netherlands. We have now removed the direct comparison of this specific threshold with Cauvery results from the Abstract, since an appropriate guideline is missing for this region.

Some of the naming of references in the text seem to be incorrect and might be caused due to the use of a reference management software. These should be checked and corrected. Examples of erroneous naming in citations include: line 38 "(Organization, 2022)" instead of World

Meteorological Organization, line 42 "(Union, 2024)" instead of Official Journal of the European Union.

**Response**: We appreciate you highlighting these inconsistencies in our reference citations. We have reviewed the manuscript thoroughly and corrected all instances of improper naming—including the specific examples you provided—to ensure accurate and formal citations in accordance with the journal's guidelines.

The authors have addressed a previous comment by the reviewers on the inclusion of ozonolysis as an atmospheric oxidative pathway for HFO-1234ze(E). They have rightfully acknowledged the temperature dependence of the ozonolysis reactions, which are not included in the McGillen et al (2023) paper (lines 47 – 54 of the revised manuscript). Moreso than the temperature dependence, one would expect OH-initiated chemistry to dominate over ozonolysis due to the reaction rates. An acknowledgement of this limitation in the production in HFC-23 from HFO-1234ze(E) would be beneficial to show why the authors have not looked into the production of HFC-23 in this paper.

**Response:** We concur with your assessment regarding the dominance of OH-initiated chemistry over ozonolysis for HFO-1234ze(E) degradation due to differences in reaction rates. We have added an explicit acknowledgment in the revised manuscript (within lines 47–57) further clarifying that formation of HFC-23 via the ozonolysis pathway represents a minor contribution relative to OH-driven removal and other loss pathways. This explanation helps to underscore why ozonolysis, though acknowledged, is not a primary focus of our quantitative analysis given its comparatively lower significance.

There is no description of how the modelled deposition data from GEOS-CHEM is superimposed onto the basin boundaries used for the watershed modelling. It would be good to include the information that was provided as a response to one of the reviewers on this topic in section 2.4

**Response:** We thank the reviewer for their feedback. This detail was presented in Section 3.2; for completeness, we have also added details of the methodology used for superimposing the atmospheric deposition data onto the watershed boundaries in Section 2.4 (Lines 245-251).

**Technical comments:**

Line 56: add in "known" between 15,000 and chemicals to read "... PFAS are a group of nearly 15,000 known chemicals...".

**Response**: The suggested edit has been incorporated. Line 60 of the revised manuscript now reads: "...PFAS are a group of nearly 15,000 known chemicals...".

Line 59: There are some PFAS compounds that take tens of decades to decompose. It might be better to show this rather than stating "many years". Line 65: Add in "known" between 2,000 and chemicals to read "... presently, there are nearly 2,000 known chemicals that have...".

**Response**: We have addressed both points. In the revised manuscript, "many years to decompose" has been revised to "tens of decades to decompose, e.g., perfluorooctanoic acid" to more accurately reflect the persistence of some PFAS compounds (Lines 61-62). The word "known" has been added, so the sentence now states: "...presently, there are nearly 2,000 known chemicals that have..." (Line 69 of the revised manuscript).

Line 107: Subtitle "Anthropogenic, Natural, and HFO-1234ze(E) emissions" does not make sense. Suggest altering to "Anthropogenic and Natural Trace gas, and HFO1234ze(E) emissions".

**Response**: We appreciate the suggestion for a more precise subtitle. The subtitle on line 111 of the revised manuscript has been updated to "Anthropogenic and Natural Trace gas, and HFO-1234ze(E) emissions" as recommended.

Lines 109 – 111: Try to avoid using double parentheses, better to list species as "... reactive gases (sulfur dioxide, SO2; nitrogen oxides, NOx; ammonia, NH3; ...". Table 1 footnotes: Adsorption spelt incorrectly on line 3 of the footnotes. Additionally, units of L/kg need to be changes to L kg-1.

**Response:** We have reviewed and corrected these formatting and textual issues. In the revised manuscript, lines 113-115, the double parentheses have been restructured to the suggested format for listing reactive gases. In Table 1, the spelling of "Adsorption" on line 3 of the footnotes has been corrected, and the units "L/kg" have been uniformly updated to "L kg-1".

Line 309, replace "Gg/yr (or kiltons/year)" with Gg yr-1 (or kilotonnes yr -1). Figure 3 units are not consistent with the rest of the paper – shown as kg/m2 • yr instead of kg m-2 yr -1.

**Response**: Thank you for pointing out these unit inconsistencies. On line 319 of the revised manuscript, "Gg/yr (or kiltons/year)" has been replaced with "Gg yr $^{-1}$  (or kilotonnes yr $^{-1}$ )". Additionally, the units in Figure 3 have been standardized to "kg m $^{-2}$  yr $^{-1}$ " for consistency with the rest of the manuscript.

Line 484: Remove "i.e.  $4.736 \times 0.04 \times 10-3$ " as this does not add any extra information to the paper than the preceding value.

**Response**: We agree that the explicit numerical breakdown was redundant. The phrase "i.e.  $4.736 \times 0.04 \times 10^{-3}$ " has been removed from line 494 of the revised manuscript, enhancing conciseness without loss of information.

Line 493 – 494: It would be pertinent to reference the limitations of this study, or at least the following section, on the results.

**Response:** We thank the Reviewer for their feedback. In the revised manuscript (Lines 503-505; Lines 519-526), we have signposted the reader to "Limitations of the Study" in the final paragraph of the "Conclusions."

Lines 506 – 510: Again, these results are based on the limitations of the study that should be acknowledged (or at least signed posted to the section on limitations) rather than stating something outright as it is misleading.

**Response**: We have addressed this by revising the statements on lines 519-526. These results are now explicitly framed within the context of the study's limitations, with direct acknowledgment and a clear signpost to the dedicated limitations section. This ensures that the conclusions are presented accurately and are not misleading.

Supplemental data: Units in the table need to be changes to Gg month-1. Additionally, the number of significant figures need to be thought about here. Captions are needed for the two plots below

the table that show propellant emissions released per month in the UK and Brazil, as well as month names.

**Response**: Thank you for the feedback on the supplemental data. We have implemented the unit change and added captions for the two plots beneath the emissions table. We have carefully reviewed and adjusted the number of significant figures across the supplementary tables to four digits, which were necessary to capture seasonal emission variations for all countries.

**Reviewer 2 comments**

This is my second assessment of the manuscript now entitled: "Atmospheric and Watershed Modelling of Trifluoroacetic Acid from Oxidation of HFO-1234ze(E) Released by Prospective Pressurized Metered-Dose Inhaler ". Although I appreciate how the authors have reacted to several of my concerns and improved the manuscript in many ways that are in line with my suggestions, I feel that my two major concerns are still not sufficiently acknowledged. Especially for the first one I don't see how the tendency of the conclusions ("no risk") can be upheld with the limited study areas.

**Response**: We thank the Reviewer for the feedback. Below, we address each comment point-by-point:

1) Generalisation of results. Again, it is concluded form the analysis of three watersheds that the TFA production from HFO-1234ze(E) emitted by pMDIs and following deposition "is minimal and does not present a risk to human health or the environment" (L33f). There is a new constraint in the same sentence ("for the regions assessed"), which is a valuable addition, but to the reader the regions assessed may still appear as being representative for maximum loads. In my view, this is not at all justified by the selection of the three watersheds. In their reply the authors claim that the watersheds were chosen because of "high pMDI emissions, which should reasonably correspond to areas of higher atmospheric TFA mass". This is a curious statement as the authors provide TFA deposition maps that clearly show that transport of HFO-1234ze(E) leads to deposition patterns that are very much different from emission patterns (compare Figures 3 and 4). I am also repeating my statement that it is also rainwater concentration rather than only deposition, which should be looked at, to assess areas with maximum impact. More TFA throughput does not mean larger concentrations. From Figures 4 and new Figure 8 I would then rather select river basins in the American Southwest as well as in Spain for the analysis instead of two only moderately affected basins (Rhine and Hudson). Although, the text does not explicitly make a universal claim anymore, the tendency is still to justify the use of HFO in pMDIs globally and, in my view, the analysed watersheds are simply not the ones one should be looking at for a worst-case scenario. At the very least section 5 and the abstract need to reflect the limitation of the obtained results in terms of global representativeness in a much clearer way before the article can get published.

**Response:** We appreciate the reviewer's emphasis on representativeness and on distinguishing emissions, deposition, and concentrations. We have revised Section 4 (Lines 519-526), Section 5 (Lines 556-564) and the title to state explicitly that our conclusions apply only to the regions and scenarios assessed, and that extension to other sectors will require additional work. We acknowledge that deposition patterns differ from emissions because transport and scavenging govern wet and dry fluxes (cf. Figs. 3–4). We also agree that concentrations depend on hydrologic dilution as well as deposition: arid regions can exhibit higher rainwater concentrations for a given atmospheric burden, yet their annual wet-deposited mass is generally lower due to limited precipitation (Vet et al., 2014).

Consistent with this, Figure 4a shows relatively higher dry deposition in parts of the American Southwest, whereas wet deposition is higher along the U.S. East Coast (Fig. 4b), reflecting greater scavenging efficiency and regional emissions (e.g., Hudson watershed). Note that, compared with

dry deposition— which tends to settle on surfaces and reach waterways indirectly via wash-off/runoff—wet deposition delivers TFA directly via rain/snow to rivers and, therefore, to surface waters. Moreover, comparing TFA rainwater concentrations between eastern and western U.S. regions (Fig. 8), we do not find a substantial difference, suggesting broadly comparable atmospheric burdens across these coasts. Wet deposition of TFA to the adjacent East Coast Ocean (Fig. 4b) is substantially higher than along the West Coast, reflecting stronger precipitation scavenging and regional transport patterns. While total pMDI use is higher in populous Eastern corridors (including the New York metropolitan area), precipitation and synoptic meteorology are the primary determinants of wet deposition.

We agree that observations in one region need not translate to others. However, based on our modelled fields, we do not find evidence that surface-water concentrations in candidate West-coast basins would be an order of magnitude higher than in the Hudson. To demonstrate the limited scope of our results, we have (i) strengthened the signposting in Section 4 (lines 521–526) regarding representativeness and study limitations; (ii) focused the abstract's quantitative comparison on the Rhine watershed, relating results to the conservative Netherlands drinking-water thresholds (as per Reviewer 1's feedback) to illustrate how our conclusions are tied to specific regions rather than global worst-case conditions; and (iii) revised the manuscript title to include "in Three Major River Basins," thereby clarifying the scope of the results.

2) Isolated view on a single use case. The authors have taken my concern up in the sense that they now compare TFA resulting from pMDI usage to TFA produced from other compounds/processes. However, the general concern that a study for an individual compound and use case concludes that there is no environmental threat, to me, is still misleading. Yes, we are not talking about the main contributor here compared to the HFOs used in refrigeration, but several small use cases will still add up and lead to an overload. I see that this is more of a philosophical question, but it should be made clear that the conclusions drawn are really isolated for the presented use case. As such I would encourage adding at least one sentence in section 5 and the abstract that reflects on this isolated view and accommodates the concerns of rising TFA levels from various usages as a whole.

Response: We agree that our assessment is intentionally limited to the pMDI use case and should not be interpreted as a statement about cumulative TFA from all sources. Our objective was to evaluate the implications of prospective HFO-1234ze(E) use in pMDIs, given its development as a replacement propellant. To make this scope explicit, we have revised Section 5 (lines 556–564) to state that a cross-sector, cumulative assessment (e.g., refrigeration/air-conditioning and other applications) is an important avenue for future work and that our conclusions apply only to the regions and scenarios analysed. We have also ensured that the title clearly restricts the scope, rather than expanding the abstract. In addition, as suggested by Reviewer 1, we underscore the specificity of our approach by reporting results for the Rhine watershed and comparing them with Netherlands drinking-water thresholds since the mouth of the Rhine river is in Netherlands.

**References**

Vet, R., Artz, R. S., Carou, S., Shaw, M., Ro, C.-U., Aas, W., Baker, A., Bowersox, V. C., Dentener, F., and Galy-Lacaux, C.: A global assessment of precipitation chemistry and deposition of sulfur, nitrogen, sea salt, base cations, organic acids, acidity and pH, and phosphorus, Atmospheric Environment, 93, 3-100, 2014.